 eLife    RESEARCH ARTICLE

# Retinal motion statistics during natural locomotion

Karl S Muller[1], Jonathan Matthis[2], Kathryn Bonnen[3], Lawrence K Cormack[1], Alex C Huk[1], Mary Hayhoe[1]*

[1]Center for Perceptual Systems, The University of Texas at Austin, Austin, United States; [2]Department of Biology, Northeastern University, Boston, United States; [3]School of Optometry, Indiana University, Bloomington, United States

**Abstract** Walking through an environment generates retinal motion, which humans rely on to perform a variety of visual tasks. Retinal motion patterns are determined by an interconnected set of factors, including gaze location, gaze stabilization, the structure of the environment, and the walker's goals. The characteristics of these motion signals have important consequences for neural organization and behavior. However, to date, there are no empirical in situ measurements of how combined eye and body movements interact with real 3D environments to shape the statistics of retinal motion signals. Here, we collect measurements of the eyes, the body, and the 3D environment during locomotion. We describe properties of the resulting retinal motion patterns. We explain how these patterns are shaped by gaze location in the world, as well as by behavior, and how they may provide a template for the way motion sensitivity and receptive field properties vary across the visual field.

## Editor's evaluation

This important study provides new information about the statistics of "retinal" motion patterns generated by human participants physically walking a straight path in real terrains that differ in ruggedness. State-of-the-art eye, head and body tracking allowed simultaneous assessment of eye movements, head movements and gait. Compelling evidence was provided for an asymmetrical gradient of flow speeds during the gait cycle of walking, tied predominantly to vertical gaze angle, together with a radial motion direction distribution tied mostly to horizontal gaze angle. This work, by describing fundamental properties of human visual motion statistics during natural behavior, should be of great interest to scientists who seek to understand the neural computations performed by walking humans, given certain behavioral goals.

*For correspondence:
hayhoe@utexas.edu

**Competing interest:** The authors declare that no competing interests exist.

## Introduction

A moving observer traveling through a stationary environment generates a pattern of motion that is commonly referred to as optic flow (*Gibson, 1950*; *Koenderink, 1986*). While optic flow is often thought of as a simple pattern of expansive motion centered on the direction of heading, this will be true for the retinal motion pattern only in the case of linear motion with gaze centered on heading direction, a condition only rarely met in natural behavior. The actual retinal motion pattern is much more complex and depends on both the three-dimensional structure of the environment and the motion of the eye through space, which in turn depends on the location of the point of gaze in the scene and the gait-induced oscillations of the body. The pervasive presence of self-motion makes it likely that the structure of motion processing systems is shaped by these patterns at both evolutionary and developmental timescales. This makes it important to understand the statistics of the actual motion patterns generated in the context of natural behavior. While much is known about

motion sensitivity in the visual pathways, it is not known how those properties are linked to behavior and how they might be shaped by experience. To do this, it is necessary to measure the actual retinal motion input in the context of natural behavior. A similar point was made by *Bonnen et al., 2020*, who demonstrated that an understanding of the retinal images resulting from binocular viewing geometry allowed a better understanding of the way that cortical neurons might encode the 3D environment.

Despite many elegant theoretical analyses of the way that observer motion generates retinal flow patterns, a detailed understanding has been limited by the difficulties in recording the visual input during locomotion in natural environments. In this article, we measure eye and body movements during locomotion in a variety of natural terrains and explore how they shape the properties of the retinal input. A number of studies have examined motion patterns generated by cameras moving through natural environments (*Betsch et al., 2005*; *Zanker and Zeil, 2005*), but these data do not accurately reflect the patterns incident on the human retinae because the movement of the cameras does not mimic the movements of the head, nor does it take into account the location of gaze. In natural locomotion, walkers gaze at different locations depending on the complexity of the terrain and the consequent need to find stable footholds (*Matthis et al., 2018*). Thus, task goals indirectly affect the motion input. In addition, natural locomotion is not linear. Instead, the head moves through a complex trajectory in space during the gait cycle, while the point of gaze remains stable in the environment, and this imparts a complex pattern of rotation and expansion on the retinal flow as recently described by *Matthis et al., 2021*. Retinal motion is generated by the compensatory rotations of the eye in space while the body moves forward during a step, and gaze is held at a fixed location in space. To characterize the properties of this motion and how it depends on gaze behavior, we simultaneously recorded gaze and image data while subjects walked in a variety of different natural terrains. In addition, to fully characterize the retinal motion we reconstructed a 3D representation of the terrain. This links the eye and body movements to the particular terrain and consequently allows calculation of the motion patterns on the retinae.

Previous work on the statistics of retinal motion by Calow and Lappe simulated retinal flow patterns using estimates of gaze location and gait oscillations, together with a database of depth images (*Calow and Lappe, 2007*; *Calow and Lappe, 2008*). However, since terrain is a profound influence on gaze deployment, the in situ data collection strategy we use here allows measurement of how gaze location varies with terrain, consequently allowing a more precise and realistic evaluation of the natural statistics than in previous studies. In this article, we focus on the interactions between gaze, body, and the resulting motion patterns. We find a stereotyped pattern of gaze behavior that emerges due to the constraints of the task, and this pattern of gaze, together with gait-induced head movements, drives much of the variation in the resulting visual motion patterns. Most importantly, because walkers stabilize gaze location in the world, the motion statistics result from the motion of the eye in space as it is carried forward by the body while counter-rotating to maintain stability. In this article, we calculate the statistics of the retinal image motion across the visual field. In addition, we describe the effects of changes in both vertical and lateral gaze angle and also the effects of natural terrain structure, independent of gaze location. Thus, a quantitative description of retinal image statistics requires an understanding of the way the body interacts with the world.

## Results

Eye movements, first-person scene video, and body movements were recorded using a Pupil Labs mobile eye tracker and a Motion Shadow full-body IMU-based capture system. Eye movements were recorded at 120 Hz. The scene camera recorded at 30 Hz with 1920 × 1080 pixel resolution and 100 deg diagonal field of view. The Shadow motion capture system recorded at 100 Hz and was used to estimate joint positions and orientations of a full 3D skeleton. Participants walked over a range of terrains two times in each direction. Examples of the terrains are shown in Figure 2a. In addition, a representation of the 3D terrain structure was reconstructed from the sequence of video images using photogrammetry, as described below in the section on optic flow estimation. Details of the procedure for calibrating and extracting integrated gaze, body, and terrain data are described in 'Methods', as well as in *Matthis et al., 2018* and *Matthis et al., 2021*.

## a.

### Saccade

### Fixation + VOR

### Eye-relative ground plane movement during VOR

## b.

### Vertical component of gaze during locomotion

Figure 1. Characteristic oculomotor behavior during locomotion. (**a**) Schematic of a saccade and subsequent gaze stabilization during locomotion when looking at the nearby ground. In the top left, the walker makes a saccade to an object further along the path. In the middle panel, the walker fixates (holds gaze) at this location for a time. The right panel shows the gaze angle becoming more normal to the ground plane during stabilization. (**b**) Excerpt of vertical gaze angle relative to gravity during a period of saccades and subsequent stabilization. As participants move forward while looking at the nearby ground, they make sequences of saccades (indicated by the gaps in the trace) to new locations, followed by fixations where gaze is held stable at a location in the world while the body moves forward along the direction of travel (indicated by the lower velocity green traces). The higher velocity saccades were detected as described in the text based on both horizontal and vertical velocity and acceleration. These are followed by slower counter-rotations of the eye in the orbit in order to maintain gaze at a fixed location in the scene (the gray time slices).

## Oculomotor patterns during locomotion

Because it is important for understanding how the retinal motion patterns are generated, we first describe the basic pattern of eye movements during locomotion as has been described previously (*Imai et al., 2001*; *Grasso et al., 1998*; *Authié et al., 2015*). *Figure 1a* shows a schematic of the typical eye movement pattern. When the terrain is complex, subjects mostly direct gaze toward the ground a few steps ahead (*Matthis et al., 2018*). This provides visual information to guide upcoming foot placement. As the body moves forward, the subject makes a sequence of saccades to locations further along the direction of travel. Following each saccade, gaze location is held approximately stable in the scene for periods of 200–300 ms so that visual information about upcoming foothold locations can be acquired while the subject moves forward during a step.

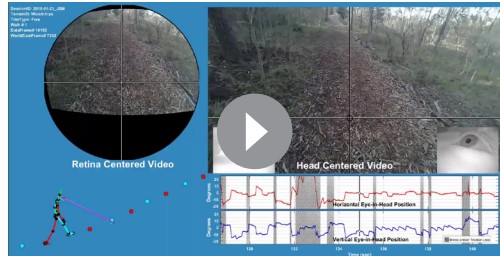

**Video 1.** Gaze behavior during locomotion. Visualization of visual input and eye and body movements during natural locomotion.
https://elifesciences.org/articles/82410/figures#video1

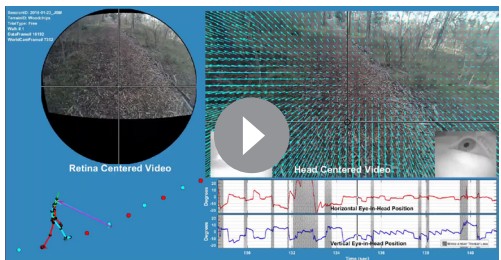

**Video 2.** Visual motion during locomotion. Visualization of eye and head centered visual motion during natural locomotion.

https://elifesciences.org/articles/82410/figures#video2

A video example of the gaze patterns during locomotion, together with the corresponding retina-centered images and traces of eye-in-head angle, is given in *Video 1*. This video is taken from *Matthis et al., 2021*, who collected a subset of the data used in this article. While the walker is fixating and holding gaze stable at a particular location on the ground, the eye rotates slowly to offset the forward motion of the body. *Figure 1b* shows an excerpt of the vertical component of gaze during this characteristic gaze pattern. Stabilization is most likely accomplished by the vestibular-ocular reflex, although other eye movement systems might also be involved. This is discussed further below and in *Matthis et al., 2021*.

During the periods when gaze location is approximately stable in the scene, the retinal image expands and rotates, depending on the direction of the eye in space, carried by the body. It is these motion patterns that we examine here. An illustration of the retinal motion patterns resulting from forward movement accompanied by gaze stabilization is shown in *Video 2*, which also shows the stark difference between retinal motion patterns and motion relative to the head. This movie is also taken from *Matthis et al., 2021*. We segmented the image into saccades and fixations using an eye-in-orbit velocity threshold of 65 deg/s and an acceleration threshold of 5 deg/s$^2$. We use the term 'fixation' here to refer to the periods of stable gaze in the world separated by saccades, 'gaze' is the direction of the eye in the scene. and gaze location is where that vector intersects the ground plane. Note that historically the term 'fixation' has been used to refer to the situation where the head is fixed and the eye is stable in the orbit. However, whenever the head is moving and gaze is fixed on a stable location in the world, the eye will rotate in the orbit. Since head movements are ubiquitous in normal vision, we use the term 'fixation' here to refer to the periods of stable gaze in the world separated by saccades, even when the eye rotates in the orbit as a consequence of stabilization mechanisms (for a review, see *Lappi, 2016* for a discussion of the issues in defining fixations in natural behavior). The velocity threshold is quite high in order to accommodate the smooth counter-rotations during stabilization. Saccadic eye movements induce considerably higher velocities, but saccadic suppression and image blur render this information less useful for locomotor guidance, and the neural mechanisms underlying motion analysis during saccades are not well understood (*McFarland et al., 2015*). We consider the retinal motion generated by saccades separately, as described in 'Methods.'.

Incomplete gaze stabilization during a fixation will add image motion. Analysis of image slippage during the fixations revealed that stabilization of gaze location in the world was very good (see Figure 10 in 'Methods'). Retinal image slippage during fixations had a mode of 0.26 deg and a median of 0.83 deg. This image slippage reflects not only incomplete stabilization but also eye-tracker noise and some small saccades misclassified as fixations, so it is most likely an overestimate. In order to simplify the analysis, we first ignore image slip during a fixation and do the analysis as if gaze were fixed at the initial location for the duration of the fixation. In 'Methods,' we evaluate the impact of this idealization and show that it is modest.

There is variation in how far ahead subjects direct gaze between terrain types, as has been observed previously (*Matthis et al., 2018*), although the pattern of saccades followed by stabilizing eye movements is conserved. We summarize this behavior by measuring the angle of gaze relative to gravity and plot gaze angle distributions for the different terrain types in *Figure 2*. Consistent with previous observations, gaze location is moved closer to the body in the more complex terrains, with the median gaze angle in rocky terrain being approximately 45 deg, about 2–3 steps ahead, and that on pavement being to far distances, a little below the horizontal. Note that the distributions are all quite broad and sensitive to changes in the terrain, such as that between a paved road and a flat dirt path. Subtle changes like this presumably affect variation in the nature of the visual information needed for foot placement. Individual subject histograms are shown in 'Methods.' There is most variability between subjects in the bark and flat terrains, as might be expected from individual

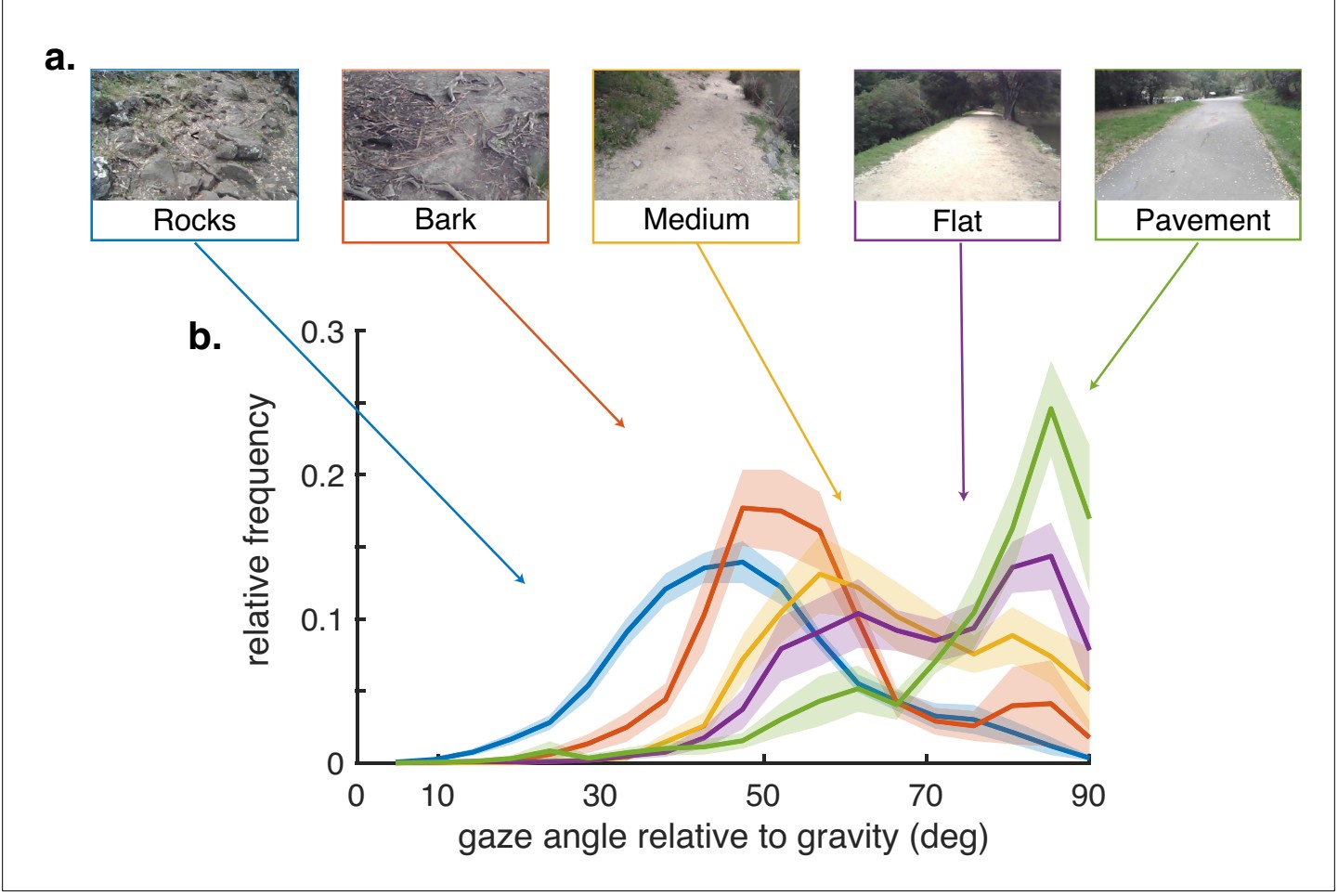

**Figure 2.** Gaze behavior depends on terrain. (**a**) Example images of the five terrain types. Sections of the hiking path were assigned to one of the five terrain types. The *Pavement* terrain included the paved parts of the hiking path, while the *Flat* terrain included the parts of the trail which were composed of flat packed earth. The *Medium* terrain had small irregularities in the path as well as loose rocks and pebbles. The *Bark* terrain (though similar to the Medium terrain) was given a separate designation as it was generally flatter than the Medium terrain but large pieces of bark and occasional tree roots were strewn across the path. Finally, the *Rocks* terrain had significant path irregularities which required attention to locate stable footholds. (**b**) Histograms of vertical gaze angle (angle relative to the direction of gravity) across different terrain types. In very flat, regular terrain (e.g. pavement, flat) participant gaze accumulates at the horizon (90°). With increasing terrain complexity participants shift gaze downward (30°–60°). Data are averaged over 10 subjects for rocky terrain and 8 subjects for the other terrains. Shaded error bars are ±1 SEM. Individual subject data are shown in 'Methods'.

trade-offs between energetic costs, stability, and other factors. The bimodality of most of the distributions reflects the observation that subjects alternate between near and far viewing, presumably for different purposes (e.g. path planning versus foothold finding). These changes in gaze angle, in conjunction with the movements of the head, have an important effect on retinal motion speeds, as will be shown below. Thus, motion input indirectly stems from behavioral goals.

## Speed and direction distributions during gaze stabilization

The way the eye moves in space during the fixations, together with gaze location in the scene, jointly determines the retinal motion patterns. Therefore, we summarize the direction and speed of the stabilizing eye movements in *Figure 3a and b*. *Figure 3a* shows the distribution of movement speeds, and *Figure 3b* shows the distribution of gaze directions (rotations of the eye in the orbit). Rotations are primarily downward as the body moves forward, with rightward and leftward components resulting from both body sway and fixations to the left or right of the future path, occasioned by the need to change direction or navigate around an obstacle. There are a small number of upward eye movements resulting from vertical gait-related motion of the head, and possibly some small saccades that were

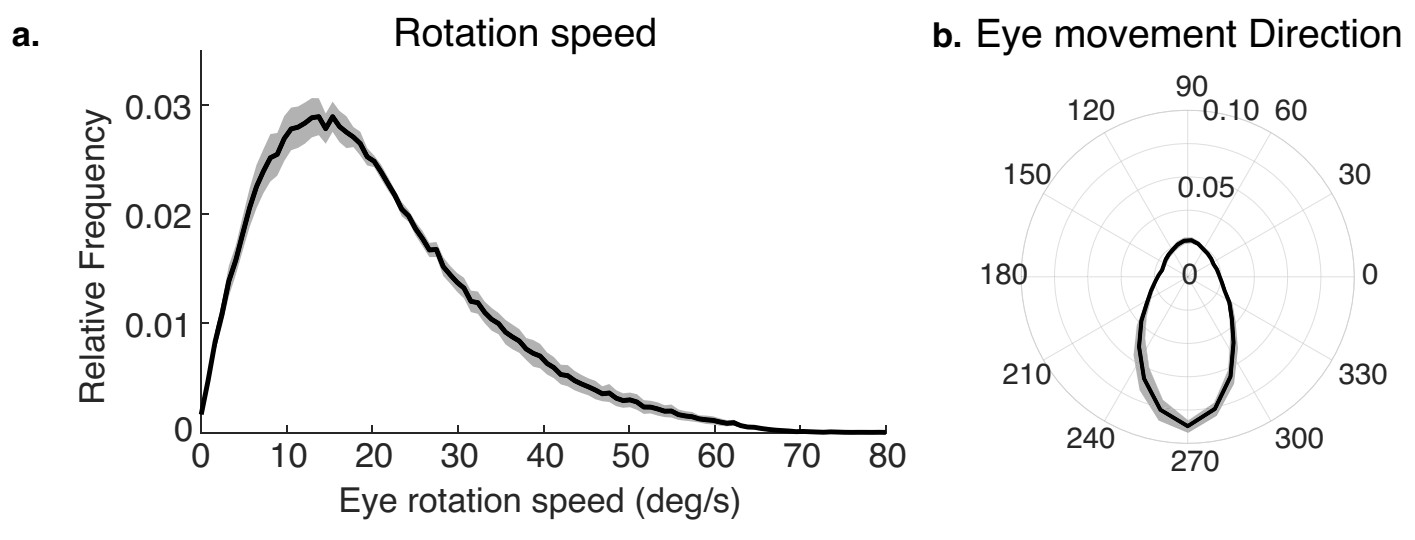

**Figure 3.** Eye rotations during stabilization. (**a**) The distribution of speeds during periods of stabilization (i.e. eye movements that keep point of gaze approximately stable in the scene). (**b**) A polar histogram of eye movement directions during these stabilizing movements. 270 deg corresponds to straight down in eye centered coordinates, while 90 deg corresponds to straight up. Stabilizing eye movements are largely in the downward direction, reflecting the forward movement of the body. Some upward eye movements occur and may be due to misclassification of small saccades or variation in head movements relative to the body. Shaded region shows ±1 SEM across 10 subjects.

misclassified as fixations. These movements, together with head trajectory and the depth structure of the terrain, determine the retinal motion. Note that individual differences in walking speed and gaze location relative to the body will affect these measurements, which are pooled over all terrains and subjects. Our goal here is simply to illustrate the general properties of the movements as the context for the generation of the retinal motion patterns.

## Optic flow estimation

In order to approximate retinal motion input to the visual system, we first use a photogrammetry package called Meshroom to estimate a 3D triangle mesh representation of the terrain structure, as well as a 3D trajectory through the terrain using the head camera video images. Using Blender (*Blender Online Community, 2021*), the 3D triangle mesh representations of the terrain are combined with the spatially aligned eye position and direction data. A virtual camera is then placed at the eye location and oriented in the same direction as the eye, and a depth image is acquired using Blender's built in z-buffer method. Thus, the depth image input at each frame of the recording is computed. These depth values per location on the virtual imaging surface are mapped to retinal coordinates based on their positions relative to the principal point of the camera. Thus, approximate depth at each location in visual space is known. Visual motion in eye coordinates can then be computed by tracking the movement of projections of 3D locations in the environment onto an image plane orthogonal to gaze, resulting from translation and rotation of the eye (see *Longuet-Higgins and Prazdny, 1980* for generalized approach).

The retinal motion signal is represented as a 2D grid where grid points $(x, y)$, correspond to polar retinal coordinates $(\theta, \phi)$ by the relationship

$$\theta = atan2(y, x)$$

$$\phi = \sqrt{x^2 + y^2}$$

Thus, eccentricity in visual angle is mapped linearly to the image plane as a distance from the point of gaze. At each $(x, y)$ coordinate, there is a corresponding speed in $\frac{deg}{s}$ and direction $atan2(\Delta x, \Delta y)$ of movement.

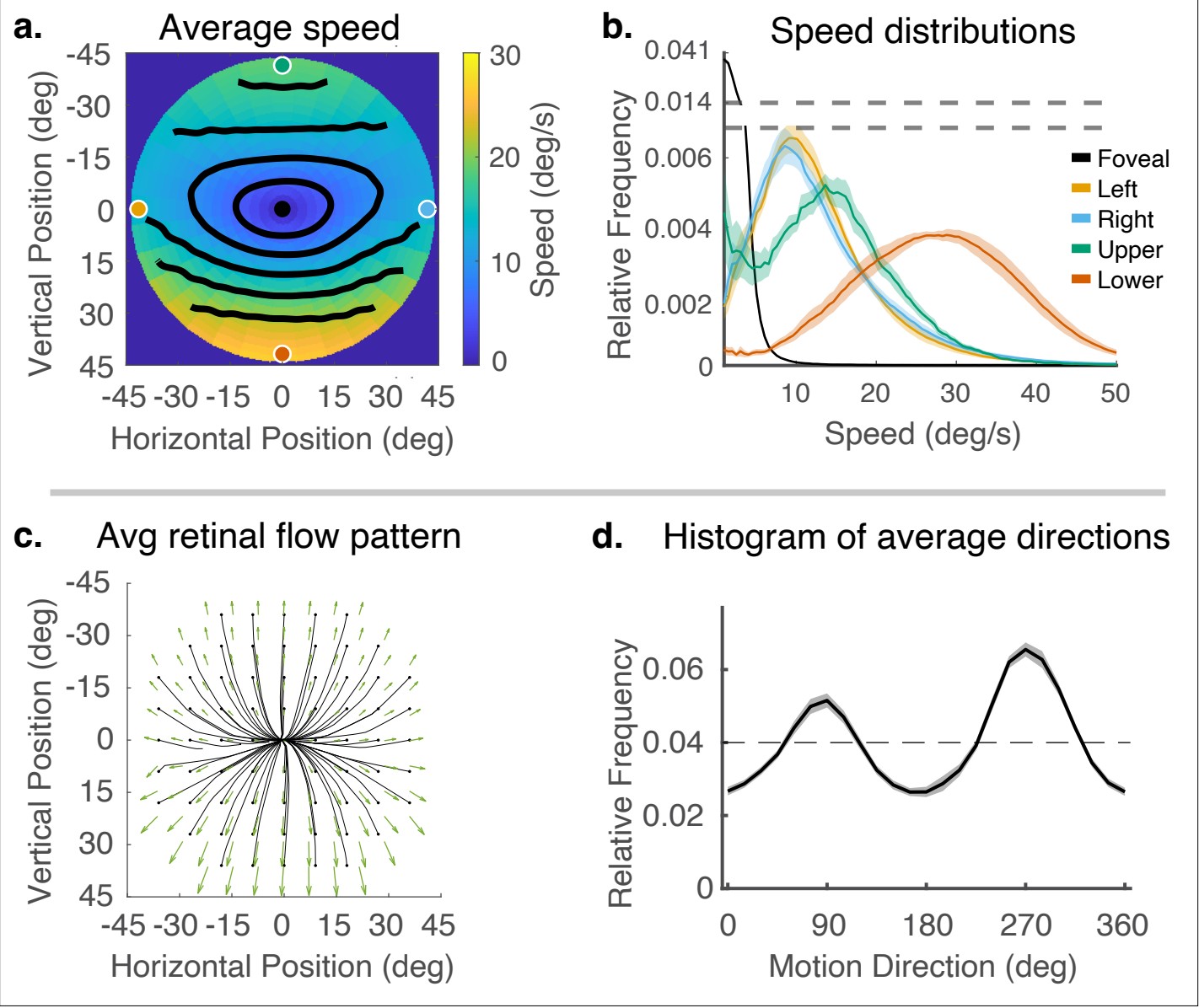

**Figure 4.** Speed and direction of retinal motion signals as a function of retinal position. (**a**) Average speed of retinal motion signals as a function of retinal position. Speed is color mapped (blue = slow, red = fast). The average is computed across all subjects and terrain types. Speed is computed in degrees of visual angle per second. (**b**) Speed distributions at five points in the visual field at the fovea and four cardinal locations. The modal speed increases in all four cardinal locations, though more prominently in the upper/lower visual fields. Speed variability also increases in the periphery in comparable ways. (**c**) Average retinal flow pattern as a function of retinal position. The panel shows the integral curves of the flow field (black) and retinal flow vectors (green). Direction is indicated by the angle of the streamline drawn at particular location. Vector direction corresponds to the direction in a 2D projection of visual space, where eccentricity from the direction of gaze in degrees is mapped linearly to distance in polar coordinates in the 2D projection plane. (**d**) Histogram of the average retinal motion directions (in **c**) as a function of polar angle. Error bars in (**b**) and (**d**) are ±1 SEM over 9 subjects.

## Average motion speed and direction statistics

Subjects' gaze angle modulates the pattern of retinal motion because of the planar structure of the environment (*Koenderink and van Doorn, 1976*). However, we first consider the average motion signal across all the different terrain types and gaze angles. We will then explore the effects of gaze angle and terrain more directly. The mean flow fields for speed and direction, averaged across subjects, for all terrains, are shown in *Figure 4*. While there will be inevitable differences between subjects caused by the different geometry as a result of different subject heights and idiosyncratic gait

patterns, we have chosen to first average the data across subjects since the current goal is to describe the general properties of the flow patterns resulting from natural locomotion across a ground plane. Individual subject data are shown in 'Methods.'.

*Figure 4a* shows a map of the average speed at each visual field location (speed is color mapped with blue being the lowest velocity and yellow being the highest, and the contour lines indicate equal speed). This visualization demonstrates the low speeds near the fovea with increasing speed as a function of eccentricity, a consequence of gaze stabilization. Both the mean and variance of the distributions increase with eccentricity as shown by the speed distributions in *Figure 4b*. The increase is not radially symmetric. The lower visual field has steeper increase as a function of eccentricity compared to the upper visual field. This is a consequence of the increasing visual angle of the ground plane close to the walker. The left and right visual field speeds are even lower than the upper visual field since the ground plane rotates in depth around a horizontal axis defined by the fixation point (see *Figure 2*). Average speeds in the lower visual field peak at approximately 28.8 deg/s (at 45 deg eccentricity), whereas the upper peaks at 13.6 deg/s.

Retinal motion directions in *Figure 4c* are represented by unit vectors. The average directions of flow exhibit a radially expansive pattern as expected from the viewing geometry. However, the expansive motion (directly away from center) is not radially symmetric. Directions are biased toward the vertical, with only a narrow band in the left and right visual field exhibiting leftward or rightward motion. This can be seen in the histogram in *Figure 4d*, which peaks at 90 deg and 270 deg. Again, this pattern results from a combination of the forward motion, the rotation in depth of the ground plane around the horizontal axis defined by the fixation point, and the increasing visual angle of the ground plane.

## Effects of horizontal and vertical gaze angle on motion patterns

Averaging the data across the different terrains does not accurately reflect the average motion signals a walker might be exposed to in general as it is weighted by the amount of time the walker spends in different terrains. It also obscures the effect of gaze angle in the different terrains. Similarly, averaging over the gait cycle obscures the effect of the changing angle between the eye and the head direction in space as the body moves laterally during a normal step. We therefore divided the data by gaze angle to reveal the effects of varying horizontal and vertical gaze angle. Vertical gaze angle, the angle of gaze in world coordinates relative to gravity, is driven by different terrain demands that cause the subject to direct gaze closer or further from the body. Vertical gaze angles were binned between 60 and 90 deg, and between 17 and 45 deg. This reflects the top and bottom third of the distribution of vertical gaze angles. We did not calculate separate plots for individual subjects in this figure as the goal is to show the kind and approximate magnitude of the transformation imposed by horizontal and vertical eye rotations.

The effect of the vertical component of gaze angle can be seen in *Figure 5*. As gaze is directed more toward the horizon, the pattern of increasing speed as a function of eccentricity becomes more radially asymmetric, with the peak velocity ranging from less than 5 deg/s in the upper visual field to speeds in the range of 20–40 deg/s in the lower visual fields. (Compare top and bottom panels of *Figure 4a and b*.) This pattern may be the most frequent one experienced by walkers to the extent that smooth terrains are most common (see the distributions of gaze angles for flat and pavement terrain in *Figure 2*). As gaze is lowered to the ground, these peaks move closer together, the variance of the distributions increase in the upper/left/right fields, and the distribution of motion speeds becomes more radially symmetric. There is some effect on the spatial pattern of motion direction as well, with the density of downward motion vectors increasing at gaze angles closer to the vertical.

Horizontal gaze angle is defined relative to the direction of travel. For a particular frame of the recording, the head velocity vector projected into a horizontal plane normal to gravity is treated as 0 deg, and the angle relative to this vector of the gaze angle projected into the same plane is the horizontal angle (clockwise being positive when viewed from above). Horizontal gaze angle changes stem both from looks off the path, and from the lateral movement of the body during a step. Body sway accounts for about ±12 deg of rotation of the eye in the orbit. Fixations to the right and left of the travel path deviate by about ±30 deg of visual angle. Data for all subjects were binned for horizontal gaze angles between –180 to –28 deg and from +28 to +180 deg. These bins represent the top and bottom eighths of the distribution of horizontal gaze angles. The effect of these changes can be seen

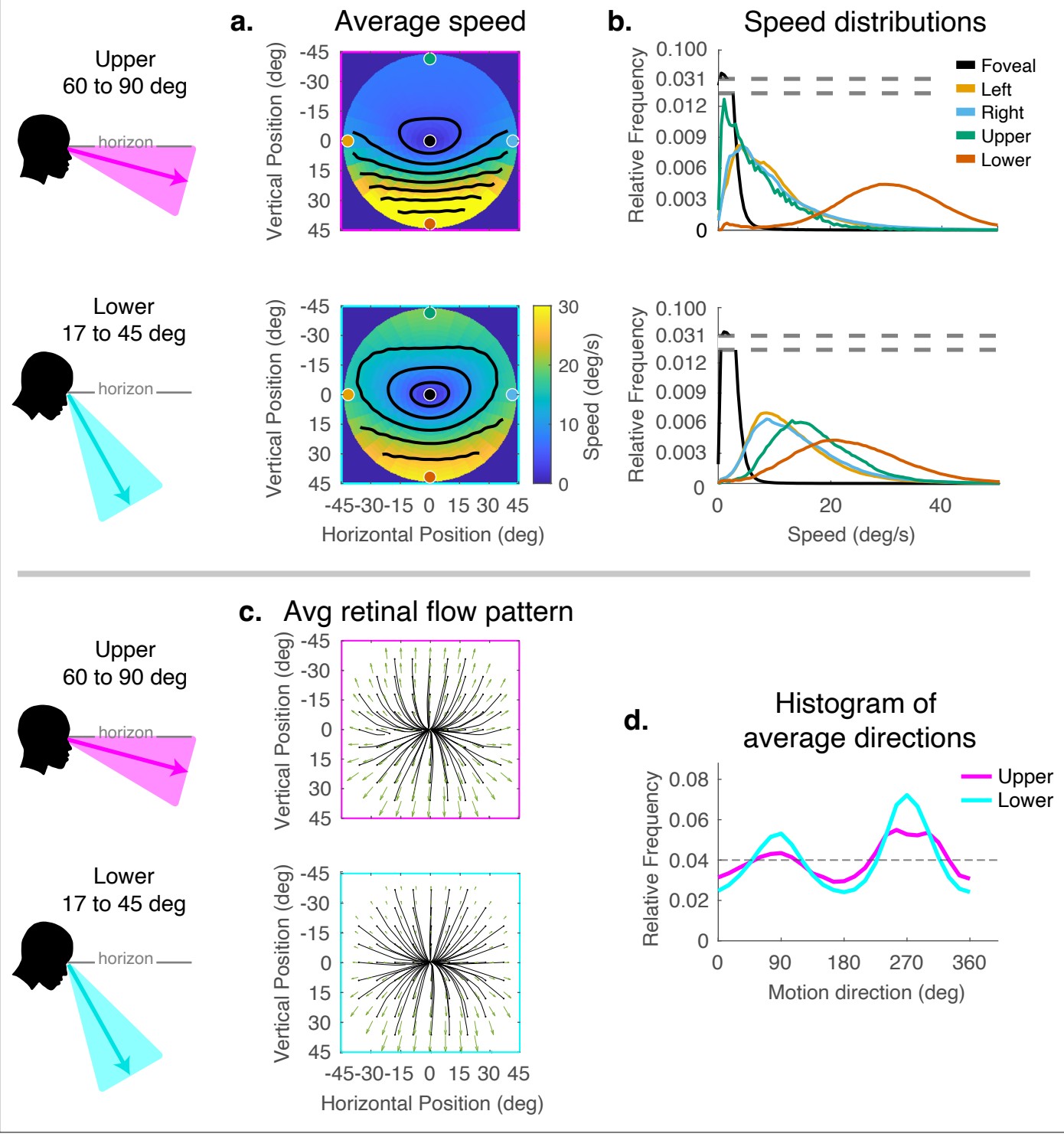

**Figure 5.** Effect of vertical gaze angle on retinal motion speed and direction. This analysis compares the retinal motion statistics for upper (60°–90°) vs. lower vertical gaze angles (17°–45°). The upper vertical gaze angles correspond to far fixations while the lower vertical gaze angles correspond to fixations closer to the body. (**a**) Average motion speeds across the visual field. (**b**) Five example distributions are shown as in *Figure 4*. Looking at the ground near the body (i.e. lower vertical gaze angles) reduces the asymmetry between upper and lower visual fields. Peak speeds in the lower visual field are reduced, while speeds are increased in the upper visual field. (**c**) Average retinal flow patterns for upper and lower vertical gaze angles. (**d**) Histograms of the average directions plotted in (**c**). While still peaking for vertical directions, the distribution of directions becomes more uniform as walkers look to more distant locations. Data are pooled across subjects.

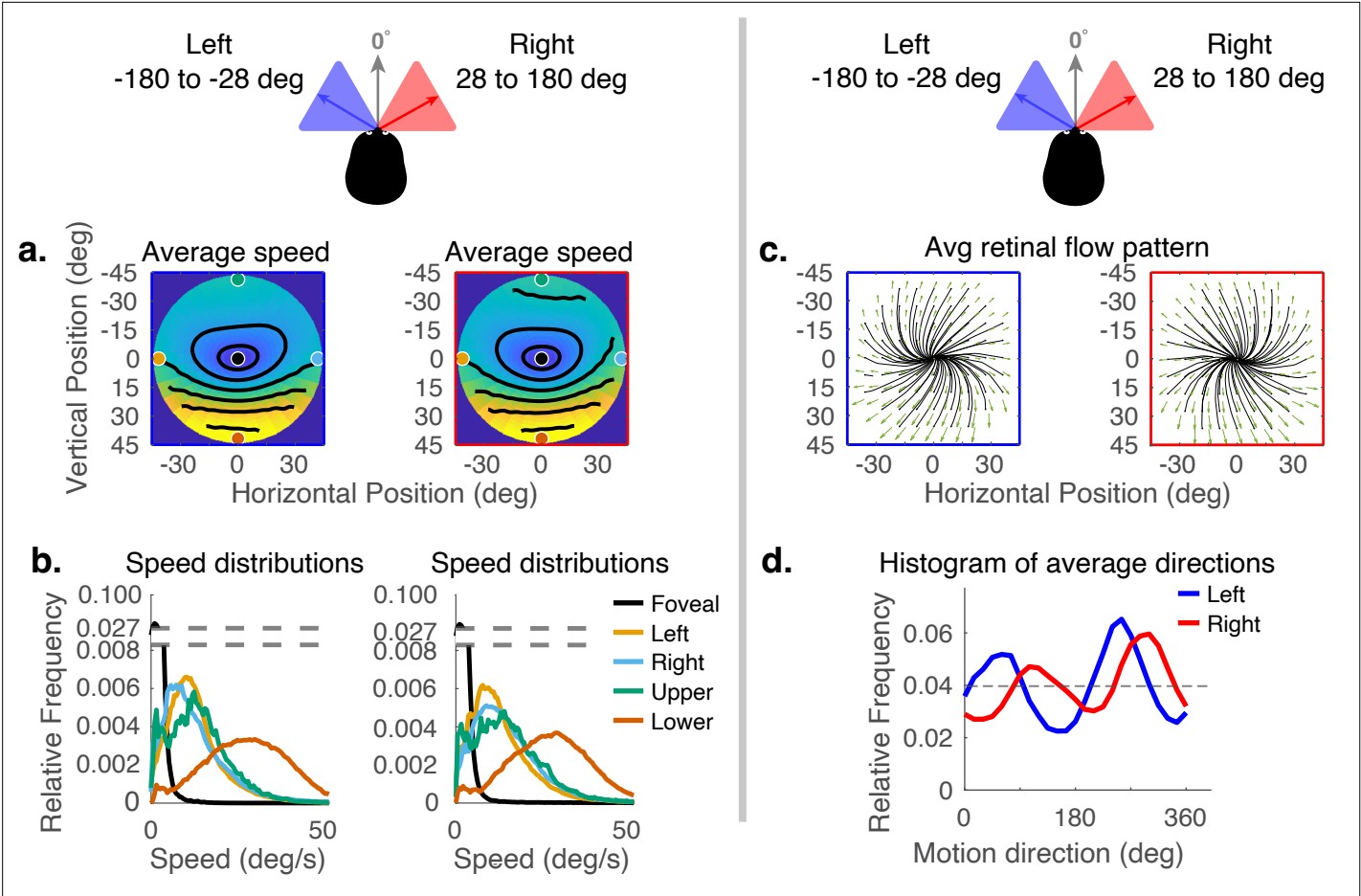

**Figure 6.** Effect of horizontal gaze angle on motion speed and direction. Horizontal gaze angle is measured relative to the head translation direction. (**a**) Average retinal motion speeds across the visual field. (**b**) Five distributions sampled at different points in the visual field, as in *Figure 4*. The effect of horizontal gaze angle on retinal motion directions is unremarkable, except for a slight tilt in to contour lines. (**c**) Average retinal flow patterns for leftward and rightward gaze angles. (**d**) Histograms of the average directions plotted in (**c**). These histograms demonstrate the shift of the rotational component of the flow field. Data are pooled across subjects.

in *Figure 6*. Changes in speed distributions are shown on the left (*Figure 6a and b*). The main effect is the tilt of the equal speed contour lines in opposite directions, although speed distributions at the five example locations are not affected very much. Changes in horizontal angle primarily influence the spatial pattern of motion direction. This can be seen in the right side of *Figure 6* (in c and d), where rightward or leftward gaze introduces clockwise or counterclockwise rotation in addition to expansion. This makes motion directions more perpendicular to the radial direction of the retinal location of the motion as gaze becomes more eccentric relative to the translation direction. This corresponds to the curl signal introduced by the lateral sway of the body during locomotion or by fixations off the path (*Matthis et al., 2021*). An example of this pattern can be seen in *Matthis et al., 2021*.

## Terrain effects on motion (independent of vertical gaze angle)

Since subjects generally look close to the body in rough terrain, and to more distant locations in smooth terrain, the data presented thus far confound the effects of 3D terrain structure with the effects of gaze angle. To evaluate the way the terrain itself influenced speed distributions, while controlling for gaze angle, we sampled from the rough and flat terrain datasets so that the distribution of samples across vertical gaze angle were matched. Thus, the comparison of flat and rocky terrain reflects only the contribution of the terrain structure to the motion patterns, This is shown in *Figure 7*. The color maps reveal a somewhat smaller difference between upper and lower visual fields in the rocky terrain than in the flat terrain. This can be seen in the contour plots and also in the distributions shown on

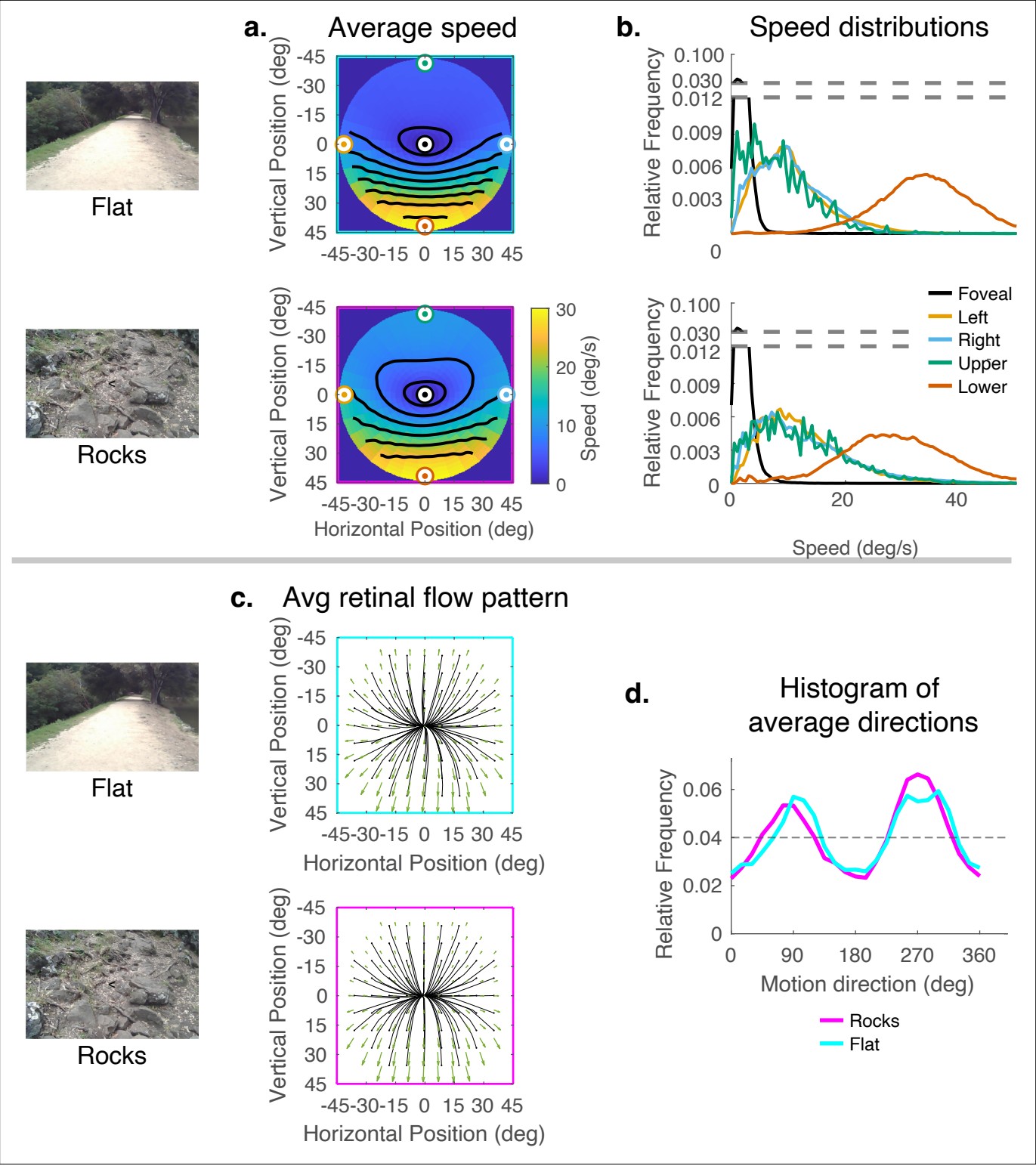

**Figure 7.** Effect of terrain (controlling for vertical gaze) on retinal motion direction and speed. While the vertical field asymmetry is slightly greater for the flat terrain, the effects of terrain on retinal motion direction and speed are modest. (**a**) Average retinal motion speeds across the visual field. (**b**) Five distributions sampled at different points in the visual field, as in *Figure 4*. (**c**) Average retinal flow patterns for leftward and rightward gaze angles. (**d**) Histograms of the average directions plotted in (**c**). Data are pooled across subjects.

the right. Thus, the added motion from the rocky terrain structure attenuated the speed difference between upper and lower visual fields, but otherwise did not affect the distributions very much. Note that the speed difference between upper and lower visual fields is greater than in *Figure 4* because gaze angle is controlled for, so the difference reflects the reduced visual angle of the distant ground plane. There is little effect on the motion direction distributions. This might be expected as the direction and speed of the motion vectors resulting from local depth variations are likely to average out over the course of the path.

## Discussion

We have summarized the retinal motion statistics experienced during locomotion in a variety of natural terrains. In these retinal motion statistics, we observe asymmetry between upper and lower visual fields, low velocities at the fovea, speeds less than 20–30 deg/s in the central 90 deg (see *Figure 4b*), the compression of directions near the horizontal meridian (see *Figure 4c*), the preponderance of vertical directions (see *Figure 4d*), modulated by lateral gaze positions (see *Figure 6d*), resulting in global rotational patterns (see *Figure 6c*). In our discussion, we will examine the relationship between these findings and existing work in retinal motion statistics, visual psychophysics, and neurophysiological measurements.

Note that this work does not (and does not intend to) sample the broad domain of motion input and pertains only to the case of locomotion in the presence of a ground plane. For example, we have not included data from self-motion in carpentered environments, where environmental structure may be close to the eye leading to higher retinal velocities, or environments containing moving objects such as pedestrians and vehicles. Specification of retinal motion in these contexts require similar databases of eye, body, and environment. Nor do the results pertain to eye movement patterns involving smooth pursuit of a moving target or optokinetic nystagmus. However, many of the features of the retinal motion distributions stem from a combination of the ground plane, the saccade and fixate strategy, the gait-induced movements of the head, and the task-linked fixation locations, so one might expect the present observations would cover a substantial domain of experienced motion. While the geometry will depend on factors such as eye height, the velocities scale with the ratio of eye height to walking speed. The results are also general across a range of normal walking speeds. The effect of gaze angle in *Figures 5 and 6*, and the effect of terrain in *Figure 7*, should allow some generalization of the data across terrains and tasks that influence the distribution of gaze locations.

While many aspects of the global pattern of motion can be understood from the geometry, the quantitative details stem from the way the body moves during the gait cycle and the way gaze behavior varies with terrain. It is therefore important to describe how these factors determine the statistics. Perhaps the most notable feature of the retinal motion is the ubiquitous pattern of very low velocities at the fovea, as expected from the pervasive stabilization of gaze. While the flow field is reminiscent of Gibson's formulation of optic flow (*Gibson, 1950*), with the focus of expansion centered at the point of gaze, the context is very different from the one Gibson first described, resulting from active stabilization rather than coincidence of gaze with the direction of travel (*Matthis et al., 2021*). Flow patterns from self-motion are generated only indirectly from body motion. The retinal motion pattern (with low velocities at the fovea) all stems from counter-rotation of the eye in the orbit to keep gaze location stable in the world, in response to the variation in momentary translation and rotation of the head during the gait cycle. This variation in momentary head direction makes head-relative flow a poor signal to guide locomotor direction, as discussed by *Matthis et al., 2021*, who suggest that instead retinal flow patterns are used for control of balance while stepping. This is likely to be important during the instabilities introduced by locomotion.

In this article, we were not concerned, however, with the how flow signals are used, or in the details of the time-varying aspect of retinal flow signals, but in the average statistics over time, which are generated by gait-induced motion. These statistics might provide a possible template for the way that properties of cortical cells or psychophysical sensitivity might vary across the visual field.

### The role of stabilizing eye movements in low foveal velocities

In a previous attempt to capture these statistics, *Calow and Lappe, 2007*; *Calow and Lappe, 2008* also found generally low velocities at the fovea, but they estimated that the gain of the stabilizing

rotations was Gaussian with a mean and standard deviation of 0.5 based on pursuit and optokinetic movements. It seems likely that the Vestibular Ocular Reflex (VOR) is the primary source of the stabilizing movements given its low latency and the increase in VOR gain during locomotion associated with spinal modulation (*Dietrich and Wuehr, 2019*; *Haggerty and King, 2018*; *MacNeilage et al., 2012*). It is possible that Optokinetic Nystagmus (OKN) and pursuit are involved as well, but the lower latency of these movements suggest the need for a predictive component. Calow and Lappe's assumption of low stabilization gain introduces added motion at the fovea and throughout the visual field. Our measurements, and previous ones (*Imai et al., 2001*; *Authié et al., 2015*), indicate that gain in real-world conditions is much closer to unity (1.0), with only modest retinal slip during fixations (10). Given the ubiquitous nature of this kind of stabilizing behavior and the pervasive presence of head movements in natural behavior, the low velocities in the central visual field should be the norm (*Land, 2019*). Consequently, gaze patterns always consisted of a series of brief fixations and small saccades to keep gaze a few steps ahead of the walker. This result supports the assumption of a motion prior centered on zero, proposed by *Weiss et al., 2002*, in order to account for particular motion illusions. Other retinal motion patterns would be experienced during smooth pursuit of a moving target, where there would be motion of the background at the fovea. The geometry will also be different for situations that evoke an optokinetic response, or situations where stabilization at the fovea may be less effective, and our results do not pertain to these situations.

## Visual field asymmetry and the effects of lateral gaze

We observed a strong asymmetry in retinal motion speed between the upper and lower visual fields (see *Figure 4c*). Further analysis of the modulation due to gaze angle reveals that this asymmetry is stronger for optic flow during gaze ahead (gaze angle of 60–90 deg, see top panel in *Figure 5b*) compared to when participants direct their gaze lower in their visual field (gaze angle of 17–45 deg, see lower panel in *Figure 5b*).

The asymmetry between upper and lower visual fields was also observed by *Calow and Lappe, 2007* and *Calow and Lappe, 2008* (in particular, see Figure 2 from *Calow and Lappe, 2007*). They concluded that these asymmetries in flow are driven primarily by the asymmetries in depth statistics. Our findings agree with this conclusion. When the gaze is lower (17–45 deg; see lower panel of *Figure 5b*), the depth statistics become more uniform between the upper and lower visual field, resulting in a much smaller difference between the retinal motion speeds in the upper vs. lower visual field. In *Calow and Lappe, 2007*, they relied on a gaze distribution that was not linked to the visual demands of locomotion, so the entirety of their optic flow statistics were examined for gaze vectors between elevations of 70–110 deg. This difference is critical. During visually guided locomotion, participants often lower their gaze to the ground. That lowering of gaze dramatically changes the relative depth statistics of the upper/lower visual field, diminishing the field asymmetries. We also note that this difference in gaze distributions (as well as the difference in VOR gain discussed above) is a big driver of why our retinal motion statistics are somewhat larger in magnitude than those reported in *Calow and Lappe, 2007*.

Additionally, our data revealed a relationship between retinal speed and direction. Specifically, we found a band of low speeds in the horizontal direction centered on the fovea. *Calow and Lappe, 2007* report no relationships between speed and direction. This difference likely stems from the task-linked fixation locations in this study. *Calow and Lappe, 2007* and *Calow and Lappe, 2008* estimated gaze locations from data where subjects viewed images of scenes or walked in different environments. In both cases, the fixation distributions were not linked to the depth images in a way that depended on immediate locomotor demands. In particular, the rugged terrains we used demanded fixations lower in the visual field to determine footholds, and this affects the angular rotation of the ground plane in pitch around the axis defined by the fixation position (see *Figure 1*). The ground fixations also underlie the preponderance of vertical directions across the visual field (see *Figure 4*). These fixations on the ground close to the body reduce the speed asymmetry between upper and lower visual fields. Added to this, we found that the motion parallax resulting from the rugged terrain structure itself reduces the asymmetry, independently of gaze angle. Thus, the behavioral goals and their interaction with body movements and terrain need to be taken into account for a complete description of retinal motion statistics.

## Retinal motion statistics and neural processing

The variations of motion speed and direction across the visual field have implications for the interpretation of the behavior of neurons in motion sensitive areas. Cells in primary visual cortex and even the retina are sensitive to motion, although it is not clear how sensitive these neural substrates might be to environmental statistics. *Beyeler et al., 2016* have argued that selectivity in the motion pathway, in particular, heading direction tuning in MSTd neurons, could simply be an emergent property of efficient coding of the statistics of the input stimulus, without the specific optimization for decoding heading direction, a function most commonly associated with MSTd. In more recent work, Mineault et al. trained an artificial neural network to relate head direction to flow patterns and found units with properties consistent with those of cells in the dorsal visual pathways. However, neither of *Beyeler et al., 2016* or *Mineault et al., 2021* directly addressed factors such as gait or gaze location relative to the ground plane during walking. These have a significant impact on the both the global patterns of retinal motion across the visual field (which we have discussed here) as well as the time-varying properties of retinal motion. Indeed, simple statistical summaries of input signals and the associated behaviors, such as gaze angle, might prove useful in understanding the responses of motion sensitive neurons if response properties arise as a consequence of efficient coding.

While these theoretical attempts to predict properties of motion sensitive pathways are encouraging, it is unclear to what extent known properties of cells in motion sensitive areas are consistent with the observed statistics. In an early paper, *Albright, 1989* observed a centrifugal directionality bias at greater eccentricities in MT, as might be expected from optic flow generated by forward movement, and speculated that this might reflect cortical sensitivity to the statistics of visual motion. While this is in principle consistent with our results, the neurophysiological data do not provide enough points of comparison to tell whether the motion sensitivity across the visual field is constrained by the current estimates of the statistics either because the stimuli to not resemble natural motion patterns or because responses are not mapped across the visual field.

Given the ubiquitous presence of a ground plane, one might expect that the preferred speed and direction of MT neurons resemble that pattern across the visual field as shown *Figure 4* with low speed preferences near the fovea, and increasing speed preferences with eccentricity. The asymmetry between speed preferences in upper and lower visual fields and the preponderance of vertical motion direction preferences (as seen in *Figure 4*) might also be observed in motion-sensitive areas as well as other features of the statistics we observe here such as the flattening of the equal speed contours. In MT, *Maunsell and Van Essen, 1987* found asymmetry in the distribution of receptive fields in upper and lower visual fields with a retinotopic over-representation in the lower visual field that might be related to the relative prominence of high velocities in the lower visual field, or more generally, might be helpful in processing ground plane motions used for navigation. Other work, however, is inconsistent with the variations observed here. For example, Maunsell and van Essen mapped speed preferences of MT neurons as a function of eccentricity out to 24 deg. While there was a weak tendency for preferred speed to increase with eccentricity, the relationship was more variable than expected from the statistics in *Figure 4*. *Calow and Lappe, 2008* modeled cortical motion processing based on retinal motion signals by maximizing mutual information between the retinal input and neural representations. They found general similarities between the tuning functions of cells in MT and their predictions. However, as in our work, it was difficult to make direct comparisons with the observed properties of cells in primate motion sensitive areas.

The role of MSTd in processing optic flow is well studied in both human and monkey cortex (*Duffy and Wurtz, 1991*; *Lappe et al., 1999*; *Britten, 2008*; *Gu et al., 2010*; *Morrone et al., 2000*; *Wall and Smith, 2008*), and activity in these areas appears to be linked to heading judgments. From our data, we might expect that the optimal stimuli for the large MSTd receptive fields would be those that reflected the variation with eccentricity seen in *Figure 4*. However, there is no clear consensus on the role of these regions in perception of self-motion. Many of the stimuli used in neurophysiological experiments have been inconsistent with the effects of self-motion since the region of low or zero velocity was displaced from the fovea. However, neurons respond vigorously to such stimuli, which would not be expected if the statistics of experience shape the properties of these cells. In humans, MST responds to motion patterns inconsistent with self-motion in a manner comparable to responses to consistent patterns (*Wall and Smith, 2008*). Similarly in monkeys, MST neurons respond to both kinds of stimulus patterns (*Duffy and Wurtz, 1995*; *Cottereau et al., 2017*; *Strong et al., 2017*).

A different way that the present results might have implications for neural processing is in possible effects of gaze angle on cell responses. Analysis of the effect of different horizontal and vertical gaze angles revealed a marked dependence of motion direction on horizontal gaze angle and of motion speed on vertical gaze angle. Motion directions were biased in the counterclockwise direction when subjects looked to the left of their translation direction, and clockwise when they looked to the right. The distribution of speed became more radially symmetric as subjects looked closer to their feet. These effects have implications for the role of eye position in processing optic flow, given the relationship between eye orientation and flow patterns. This relationship could be learned and exploited by the visual system. Effects of eye position on response properties of visual neurons have been extensively observed in a variety of different regions, including MT and MST (*Bremmer et al., 1997*; *Andersen et al., 1990*; *Boussaoud et al., 1998*). It is possible that such eye direction tuning could be used to compute the expected flow patterns and allow the walker to detect deviations from this pattern. *Matthis et al., 2021* suggested that the variation of the retinal flow patterns through the gait cycle could be learnt and used to monitor whether the variation in body movement was consistent with stable posture. Neurons that compute the consistency between expected and actual retinal motion have been observed in the primary visual cortex of mice, where firing rate is related to the degree of mismatch between the experimentally controlled and the anticipated motion signal given a particular movement (*Zmarz and Keller, 2016*). It is possible that a similar strategy might be employed to detect deviations from expected flow patterns resulting from postural instability.

The most closely related work on neural processing in MSTd has carefully mapped the sensitivity of neurons to translation, rotation, expansion, and their combinations, beginning with *Duffy and Wurtz, 1995*; *Graziano et al., 1994*. This work explicitly investigates stimuli that simulate a ground plane (*Lappe et al., 1996*) and encompasses the relationship of that tuning with eye movements (*Bremmer et al., 1997*; *Andersen et al., 1990*; *Boussaoud et al., 1998*; *Bremmer et al., 2010*) and tasks like steering (*Jacob and Duffy, 2015*). One of the limitations of these findings is that they rely on simulated visual cues for movement through an environment and thus result in inconsistent sensory cues. However, recent advances allow for neurophysiological measurements and eye tracking during experiments with head-fixed running, head-free, and freely moving animals (from mice to marmosets and even macaque monkeys). These emerging paradigms will allow the study of retinal optic flow processing in contexts that do not require simulated locomotion. We caution that these experiments will need to take a decidedly comparative approach as each of these species has a distinct set of body movement patterns, eye movement strategies, a different relationship to the ground-plane, and different tasks it needs perform. Our findings in this article strongly suggest that all of these factors will impact the retinal optic flow patterns that emerge, and perhaps the details of the tuning that we should expect to find in neural populations sensitive to motion. While the exact relation between the retinal motion statistics we have measured and the response properties of motion-sensitive cells remains unresolved, the emerging tools in neurophysiology and computation make similar approaches with different species more feasible.

## Conclusions

In summary, we have measured the retinal motion patterns generated when humans walked through a variety of natural terrains. The features of these motion patterns stem from the motion of the eye while walkers hold gaze stable at a location in the scene, and the retinal image expands and rotates as the body moves through the gait cycle. The ground plane imparts an increasing speed gradient from upper to lower visual field that is most likely a ubiquitous feature of natural self-generated motion. Gaze location varies with terrain structure as walkers look closer to their bodies to control foot placement in more rugged terrain, and this reduces the asymmetry between upper and lower visual fields, as does the motion resulting from the more complex 3D terrain structure. Fixation on the ground plane produces a preponderance of vertical directions, and lateral rotation of the eye relative to body direction, both during the gait cycle and while searching for suitable paths, changes the distribution of motion directions across the visual field. Thus, an understanding of the complex interplay between behavior and environment is necessary for understanding the consequences of self-motion on retinal input.

## Methods
### Experimental task and data acquisition
Processed data used to generate figures is shared via Dryad. Code to generate figures is shared via Zenodo. Raw data as well as analysis and visualization is available via Dryad.

### Participants
Two datasets were used in this study. Both were collected using the same apparatus, but from two separately conducted studies with similar experimental conditions. One group of participants (n = 3, two males, one female) was recruited with informed consent in accordance with the Institutional Review Board at the University of Texas at Austin (approval number 2006-06-0085) and collected in an Austin area rocky creek bed. The second participant group (n = 8, four males, four females) was recruited with informed consent in accordance with the Institutional Review Board at The University of California Berkeley (2011-07-3429) and collected in a nearby state park. Subjects were either normal or corrected to normal acuity, and ranged in age from 24 to 54 y.

### Equipment
Infrared eye recordings, first-person scene video, and body movements of all participants were recorded using a Pupil Labs mobile eye tracker (*Kassner et al., 2014*) combined with a Motion Shadow full-body IMU-based motion capture system (Motion Shadow, Seattle, WA). The eye tracker has two infrared eye cameras and a single outward-facing scene camera. Each eye camera records at 120 Hz at 640 × 480 resolution, while the outward-facing scene camera records at 30 Hz with 1920 × 1080 pixel resolution, with a 100 deg diagonal field of view. The scene camera is situated approximately 3 cm above the right eye. The Shadow motion capture system is comprised of 17 three-axis accelerometer, gyroscope, and magnetometer sensors. The readings from the suit are processed by software to estimate joint positions and orientations of a full 3D skeleton. The sensors record at 100 Hz, and the data is later processed using custom MATLAB code (MathWorks, Natick, MA). See *Matthis et al., 2018* and *Matthis et al., 2021* for more details.

### Experimental task
For both groups of participants (Austin dataset and Berkeley dataset), the experimental task was similar, with variation in the terrain types between the two locations. For the Berkeley participants, the task involved walking back and forth along a loosely defined hiking trail that varied in terrain features and difficulty. This trail was traversed two times in each direction by each participant. Different portions of the trail were pre-examined and designated as distinct terrain types, being labeled as one of pavement, flat (packed earth), medium (some irregularities such as rocks and roots), and bark (large pieces of bark and roots but otherwise not rocky), and rocks (large rocks that constrained the possible step locations). Examples of the terrain are shown in *Figure 2*. For the Austin participants, the task involved walking back and forth along a rocky dried out creek bed. Participants walked three times in each direction. This is the same terrain used in *Matthis et al., 2018*. This terrain difficulty is most comparable to the rocks condition in the Berkeley dataset. For each of the rocky terrains, the ground was sufficiently complex that subjects needed to use visual information in order to guide foot placement (see *Matthis et al., 2018* for more details).

### Calibration and postprocessing
At the start of each recording session, subjects were instructed to stand on a calibration mat that was used for all subjects. The calibration mat had marked foot locations that were at a fixed distance from a calibration point 1.5 m away from the foot locations. Subjects were then instructed to maintain fixation on this calibration point throughout a calibration process. The calibration process involved rotating the head while maintaining fixation, to eight different predetermined head orientations (the cardinal and oblique directions). This segment of each subjects recording was later used to align and calibrate the data. This was done by finding an optimal rotation that aligned the mobile eye tracker's coordinate system to that of the motion capture system, such that the distance between the projected gaze vector and the calibration point on the mat was minimized. This rotation was then applied to the position data in the recording, aligning the rest of the recording in space. Each system's timestamps

were then used to align the recording temporally as timestamps were recorded to a single laptop computer on a backpack worn by subjects throughout the recording. (See *Matthis et al., 2018* for more details.) The 100 Hz motion capture data was upsampled using linear interpolation to match the 120 Hz eye tracker data.

## Fixation detection

During locomotion, walkers typically maintain gaze on a stable location in the world as the body moves forward (*Matthis et al., 2018*). During these periods, the eye counter-rotates in the orbit, driven largely by the vestibuloocular reflex, although other eye movement systems might also be involved (*Matthis et al., 2021*). We refer to these periods of stable gaze in the world in the presence of a smooth compensatory eye rotation as 'fixations,' although in more common usage of the term the eye is stable in the orbit and the head is fixed. In order to analyze the retinal motion input, we needed to differentiate between saccades and periods of stable gaze since vision is suppressed during saccades, and the visual information used for guidance of locomotion is acquired during the fixation periods. We therefore needed to segment the eye position data into fixations and saccades. The presence of smooth eye movements during stable gaze makes detection of fixations more difficult. We used a velocity and acceleration threshold method with thresholds set such that detected fixations best match hand-coded fixations. The velocity threshold was 65 deg/s velocity and the acceleration threshold was 5 deg/s$^2$. Frames of the recording that fall below these thresholds are labeled as fixation frames, and sequential fixation frames are grouped accordingly into fixation instances. The velocity threshold is quite high in order to accommodate the smooth counter-rotations of the eye in the orbit during stabilization. Fixation identified by this algorithm were then checked against manual coding using the trace of velocity versus time.

## Fixation idealization via initial location pinning

Incomplete stabilization of gaze location in the world during a fixation will add image motion. Analysis of these small eye movements revealed that stabilization of gaze location in the world was very effective. The effectiveness of stabilization of gaze was determined by measuring the deviation of gaze

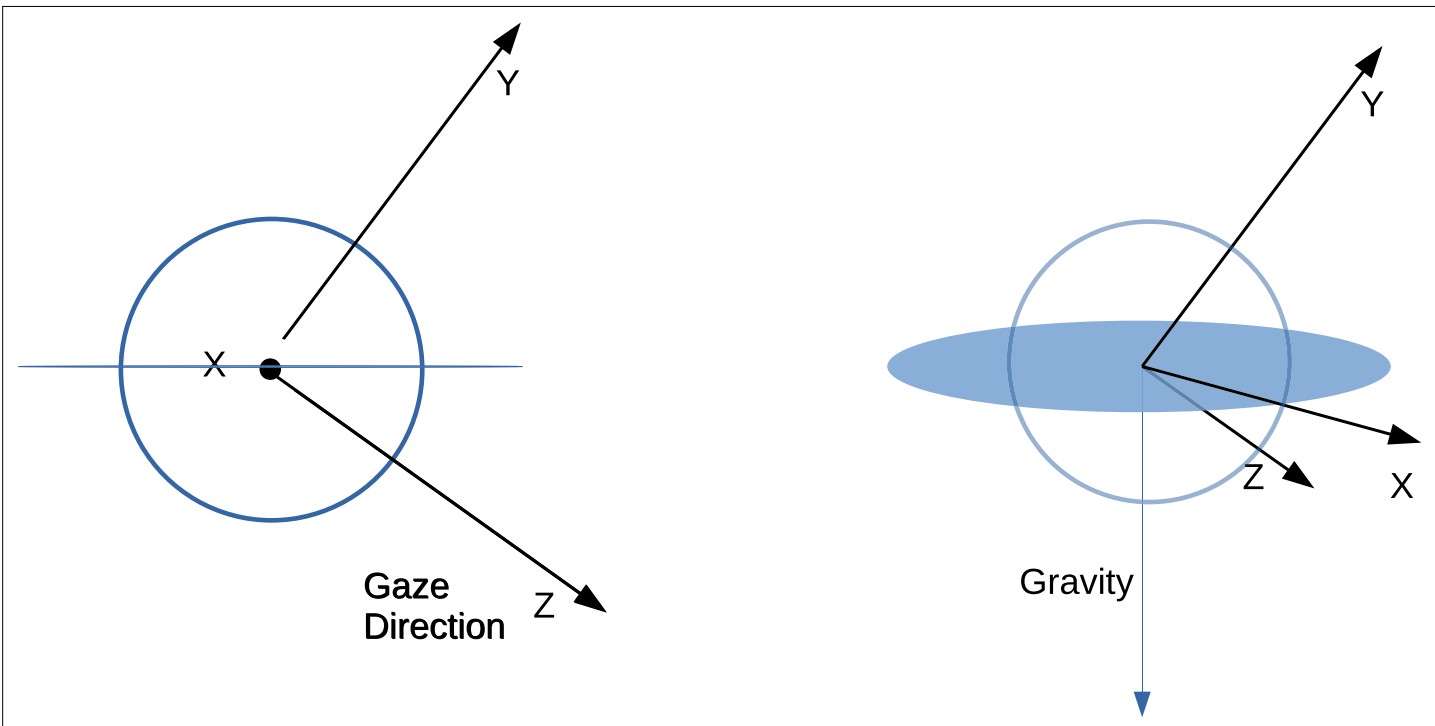

**Figure 8.** Schematic depicting eye relative coordinate system. Left and right show basis vectors used from different viewpoints. Z corresponds to the gaze direction in world coordinates. Then X is the vector perpendicular to both Z and the gravity vector. Finally, the Y coordinate is the vector perpendicular to both X and Z. The X vector thus resides within the plane perpendicular to gravity.

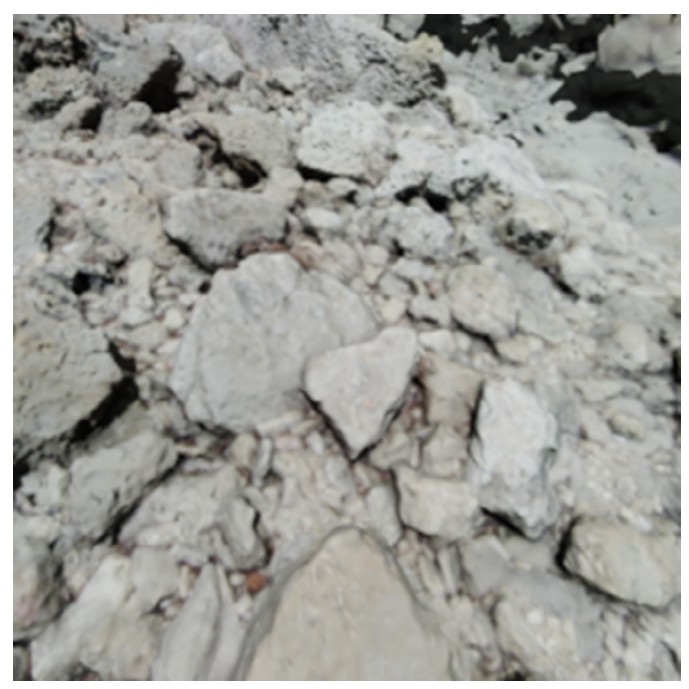

**Figure 9.** Sample of the terrain reconstruction is rendered as RGB image showing the level of detail provided by the photogrammetry.

from the initially fixated location over the duration of the fixation. This slippage had a median deviation of 0.83 deg and a mode of 0.26 deg (see below for more details). This image slippage reflects a combination of imperfect stabilization, and artifacts such as small saccades that were misclassified as fixations, and noise in eye tracker signal. In order to simplify the analysis, we first ignore image slip during a fixation and do the analysis as if gaze were fixed at the initial location for the duration of the fixation. We evaluate the impact of this idealization below. Note that in Calow and Lappe's previous studies they assumed that the stabilization gain varied randomly between 0 and 0.5, which would correspond to much less effective stabilization. This estimate was based on experiments where the subject was stationary, so mostly likely pursuit or optokinetic nystagmus were evoked by the image motion. These eye movement systems have very different drivers and properties than the vestibular-ocular reflex that is most likely responsible for the stabilizing behavior observed during locomotion. Other demonstrations of the precision of stabilization during locomotion have shown good stabilization of the foveal image on the retina (refs). Since spinal mechanisms increase the gain of the vestibular-ocular reflex during locomotion it seems most likely that VOR is the primary contributor, given its low latency, although it is possible that OKN and potentially predictive pursuit make some contribution (*Dietrich and Wuehr, 2019*; *Haggerty and King, 2018*; *MacNeilage et al., 2012*).

## Retinocentric coordinate system and eye movement directions

The first goal of the study was to compute the retinal motion statistics caused by movement of the body during the periods of (idealized) stable gaze, which we refer to here as fixations. During the periods of stable gaze, retinal motion is caused by expansion of the image as the walker takes a step forward, together with the counter-rotations of the eye in the orbit induced by gait, around the yaw axis.

Eye movement directions are computed over sequential frame pairs in the spatial reference frame of the first frame in the pair. This is done by considering sequential frame pairs within a fixation or a saccade. Then eye coordinate basis vectors are calculated for the first frame in a pair. We first define an eye relative coordinate system X, Y, Z. Gaze direction is the third dimension (orthogonal to the plane within which the eye movement direction will be calculated), the first dimension is the normalized cross-product between the eye direction and the negative gravity in world coordinates.

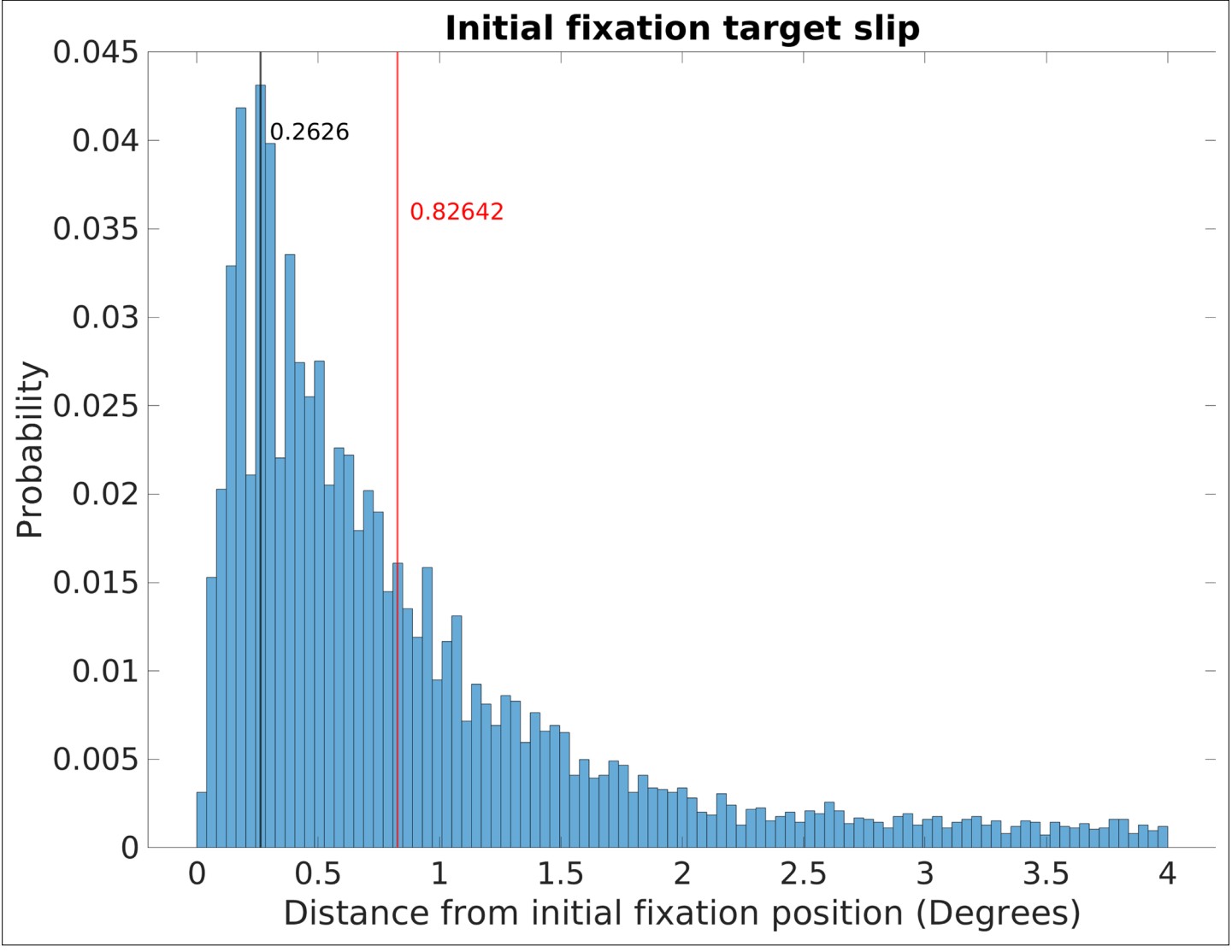

**Figure 10.** Retinal image slippage. Normalized relative frequency histogram of the deviation from the location of the initial fixation over the course of a fixation. Initial fixation location is computed and tracked over the course of each fixation, and compared to current fixation for duration of each fixation. Median deviation value is calculated for each fixation. The histogram captures the extent of variability of initially fixated locations relative to the measured location of the fovea over the course of fixations, with most initially fixated locations never deviating more than 2 deg of visual angle during a fixation.

The second dimension is the cross-product between the first and third dimensions. Thus, the basis X, Y, Z changes on each frame of the recording. This convention assumes that there is no torsion about the viewing axis of the eye, and so one dimension (the first) stays within a plane perpendicular to gravity. Eye movements and retinal motion are both represented as rotations within this coordinate system. These rotations last for one frame of the recording and are described by a start direction and end direction in X, Y, Z. Using these coordinates, the next eye direction (corresponding to the eye movement occurring over frame pairs) is computed in the reference frame of the first eye direction's coordinates $(x, y, z)$. The direction is then computed as

$$atan2(y, x)$$

A schematic depicting this coordinate system can be seen in *Figure 8*. The eye tracker combined the estimates of gaze point from the two eyes that was used as the point of fixation. We estimate retinal motion on a single cyclopean retina fixated on that point.

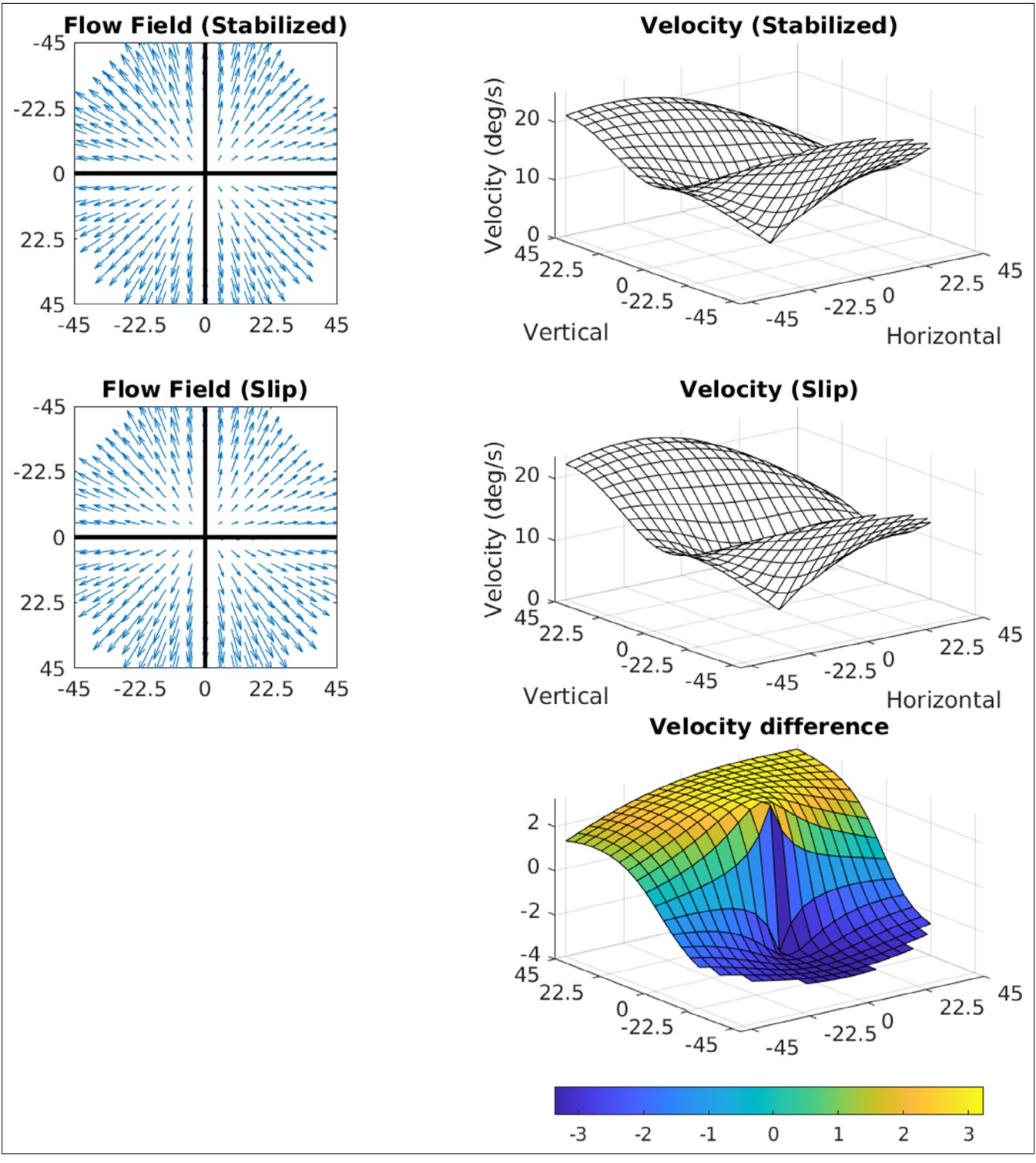

**Figure 11.** Effects of 3.2 deg/s slip on velocity. The median slip of approximately 0.8 deg during a 250 ms fixation would lead to a retinal velocity of 3.2 deg/s at the fovea. The figure shows how retinal slip of this magnitude would affect speed distributions across the retina. The flow fields in the two cases (perfect stabilization and 0.8 deg of slip) are shown on the left, and the speed distributions are shown on the right. The bottom plot shows a heat map of the difference.

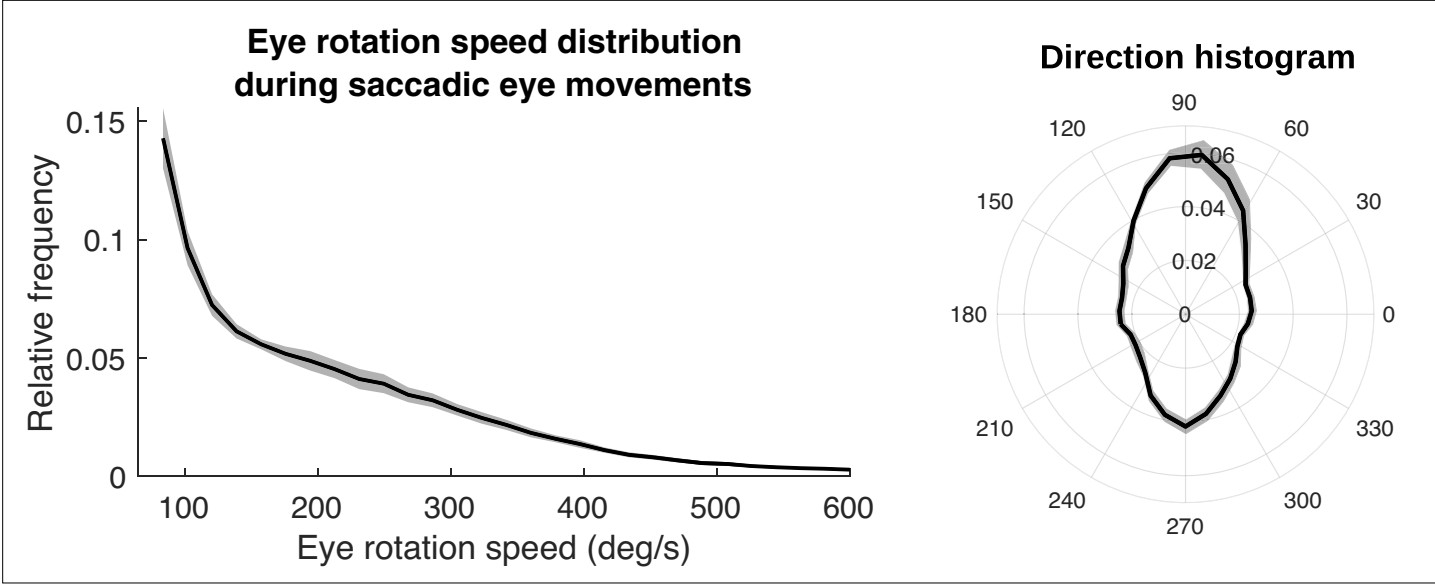

**Figure 12.** Speed and direction distributions for saccades. The distribution of angular velocities of the eye during saccadic eye movements is shown on the left. On the right is a 2D histogram of vertical and horizontal components of the saccades.

## 3D terrain structure measurement using photogrammetry

Terrain reconstructions used a computer vision method called photogrammetry, which allows simultaneous estimation of camera poses and environmental structure from sequences of camera images. This was handled using an open-source photogrammetry tool, Meshroom (*AliceVision, 2018*), which bundles various well-established algorithms for different steps in a mesh reconstruction process. The first step is to extract image features from each frame of the input sequence using the SIFT (Ref 93Karl) algorithm and then pairs of image matches are found across the image sequences, allowing calculation of the displacement across the matched images. This contains information about how the camera moved through the environment, as well as information about the structure of the environment. With image pairs selected, as well as corresponding feature locations computed, scene geometry and camera position and orientation for each image pair can then be estimated using structure from motion methods. Structure from motion yields a sparse reconstruction of the environment, represented by 3D points in space, that is used to estimate a depth for each pixel of the input images using the estimated camera pose for each image and the 3D points in view of that camera pose. After each camera view has an estimated depth map, all depth maps are fused into a global octree that is used to create a 3D triangle mesh. For the terrain reconstruction, we leveraged the 3D triangle mesh output, which contained information about the structure of the environment. Blender's z-buffer method was used in order to calculate depth maps, and these depth maps were then used to compute retinal motion inputs using geometry.

Optic flow estimation relied on 3D terrain reconstruction provided by Meshroom. Default parameters were used in Meshroom, with the exception of specifying camera intrinsics (focal length in pixels, and viewing angle in degrees). RGB image-based scene reconstruction is subject to noise that will be influenced by the movement of the camera and its distance from the terrain. In order to evaluate the accuracy of the 3D reconstruction, we took advantage of the terrain meshes calculated from different traversals of the same terrain by an individual subject and also by the different subjects. Thus, for the Austin dataset we had 12 traversals (out and back three times by two subjects). Easily identifiable features in the environment (e.g. permanent marks on rocks) were used in order to align coordinate systems from each traversal. A set of corresponding points between two sets of points can be used in order to compute a single similarity transform between those points. Then the iterative closest point method is used to align the corresponding point clouds at a finer scale by iteratively rotating and translating the point cloud such that each point moves closer to its nearest neighbor in the target point cloud. The process is repeated for each recording, and the resulting coordinate transformation is then applied to all recordings respectively such that they are all in the same coordinate frame.

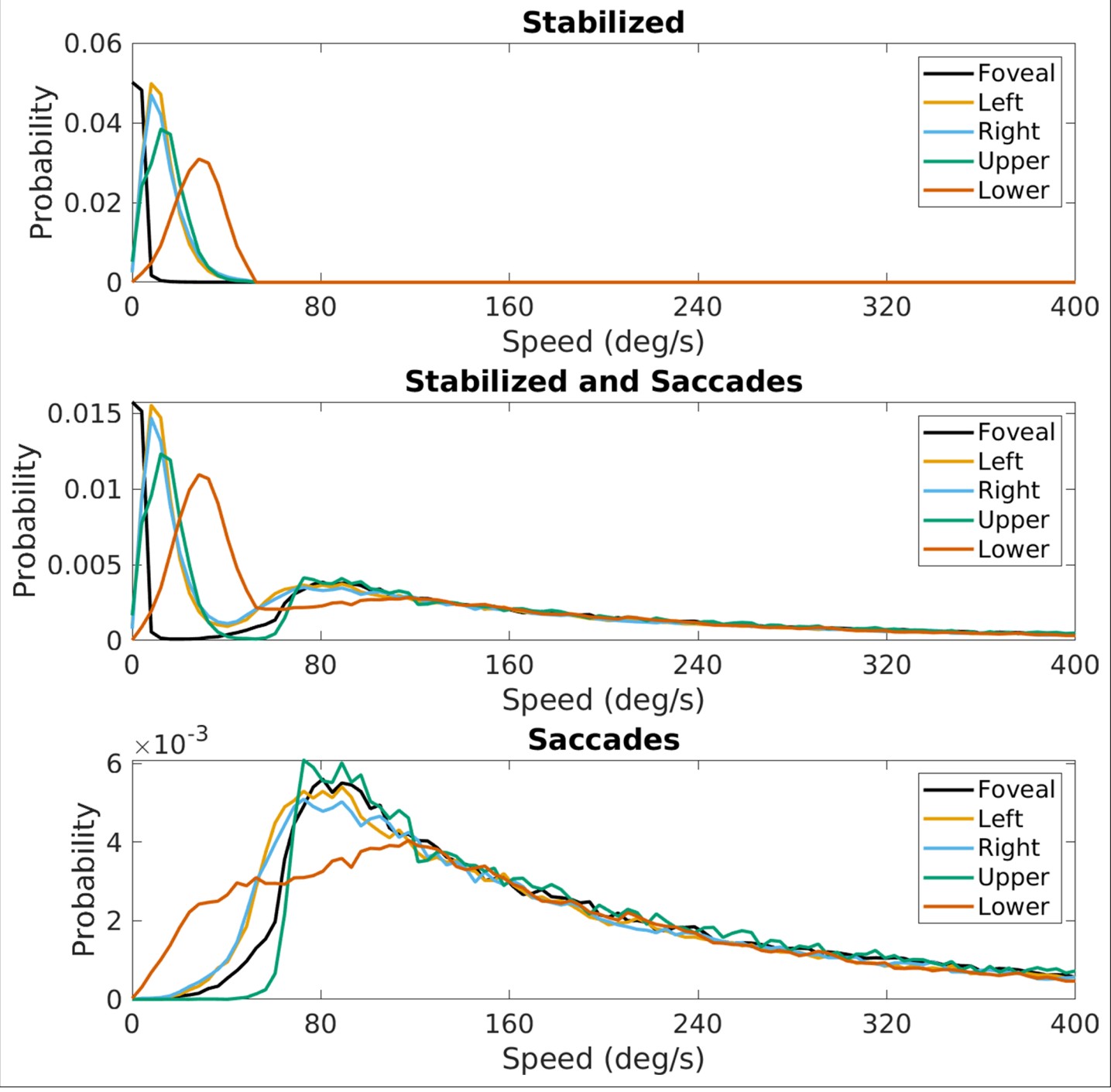

**Figure 13.** Effect of saccades on speed distributions. Speed distributions at the five locations in the visual field shown in *Figure 4*. The top panel is replotted from *Figure 4*, the middle panel shows these distributions with the saccades added, and the bottom panel shows the contribution from the saccades alone.

Foothold locations were estimated with the different aligned meshes using a method that found the closest location on a mesh to the motion capture system's foothold location measurement. The distances between corresponding foothold locations between meshes were then measured. There was high agreement between terrain reconstructions, with a median distance of approximately 3 cm, corresponding to 0.5 deg of visual angle. Thus, the 3D structure obtained is quite consistent between different reconstructions of the same terrain. An example of the terrain reconstruction can be seen

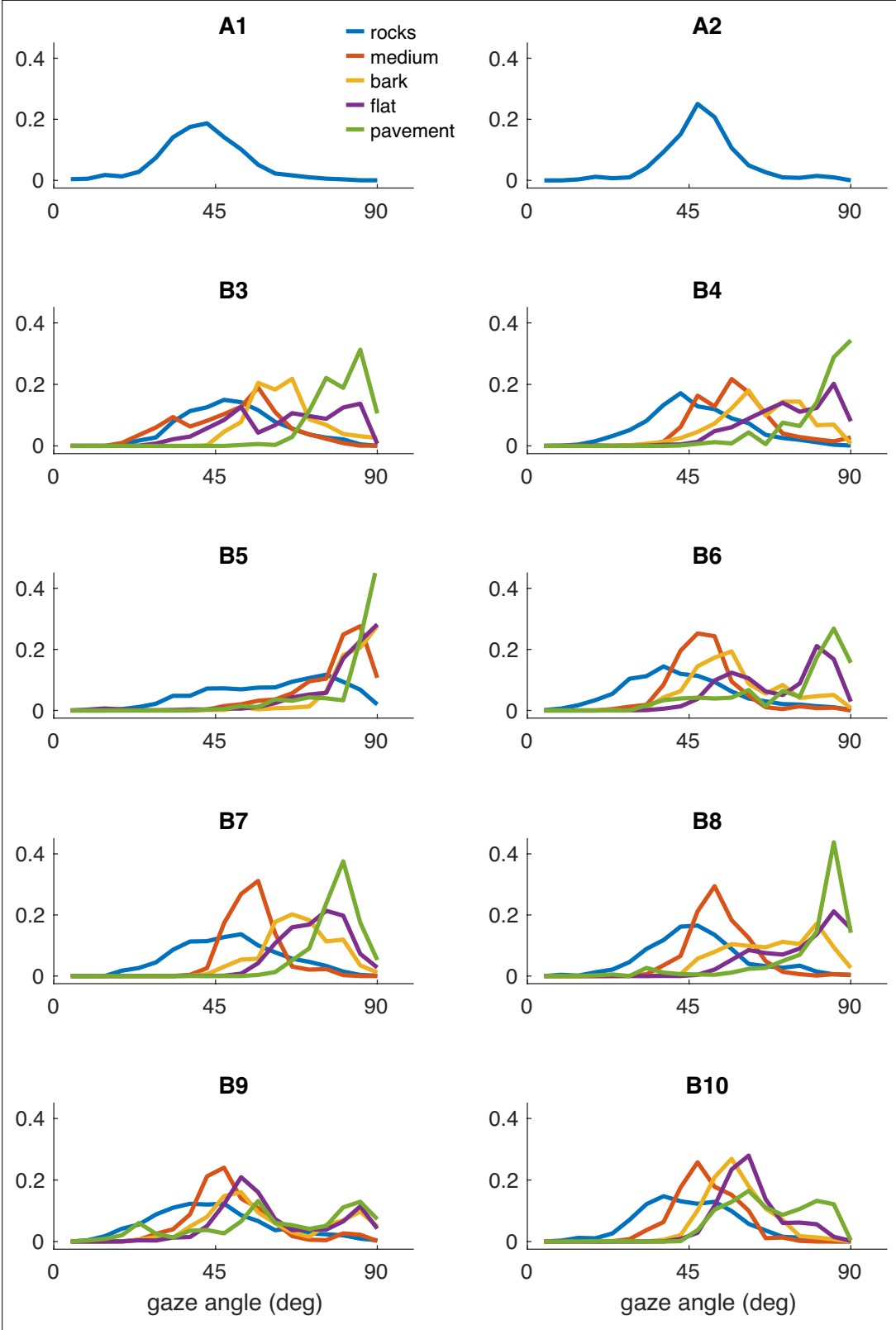

**Figure 14.** Individual participant data – compare to *Figure 2*. Histograms of vertical gaze angle (angle relative to the direction of gravity) across different terrain types. In very flat, regular terrain (e.g. pavement, flat) participant gaze accumulates at the horizon (90°). With increasing terrain complexity, participants shift gaze downward (30°–60°). A total of 10 participants are shown. Participants A1–2 correspond to two participants whose data was collected in Austin, TX. The data from these participants includes only the 'rocks' terrain. B3–B10 correspond to eight participants whose data was collected in Berkeley, CA.

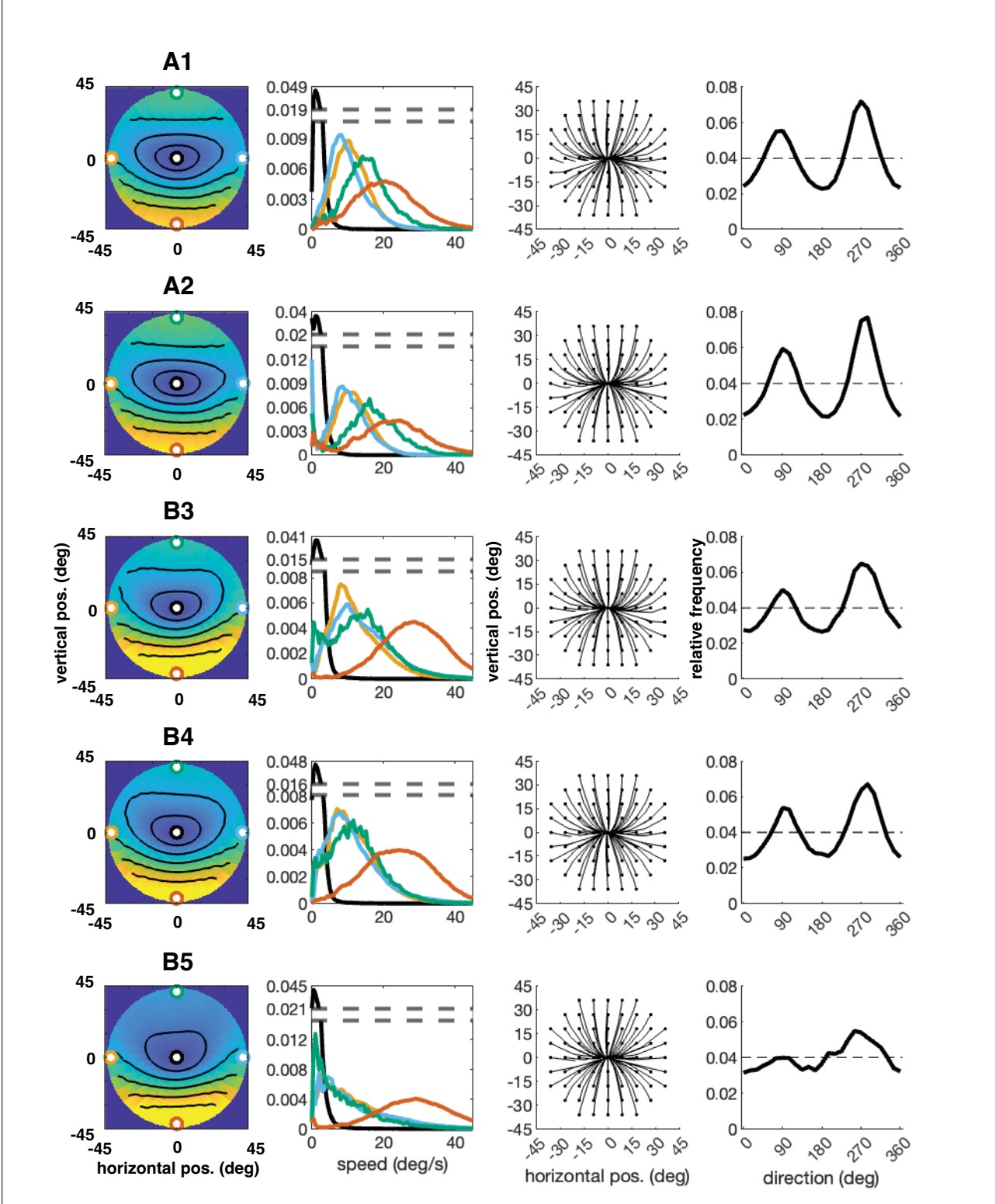

**Figure 15.** Individual participant data – compare to *Figure 4* (five participants: A1–A2, B3–B5). Column 1: average speed of retinal motion signals as a function of retinal position. Speed is color mapped (blue = slow, red = fast). The average is computed across all subjects and terrain types. Speed is computed in degrees of visual angle per second. Column 2: speed distributions at five points in the visual field (the fovea and four cardinal locations). Column 3: average retinal flow pattern. Column 4: Histogram of the average retinal motion directions (Column 3) as a function of polar angle.

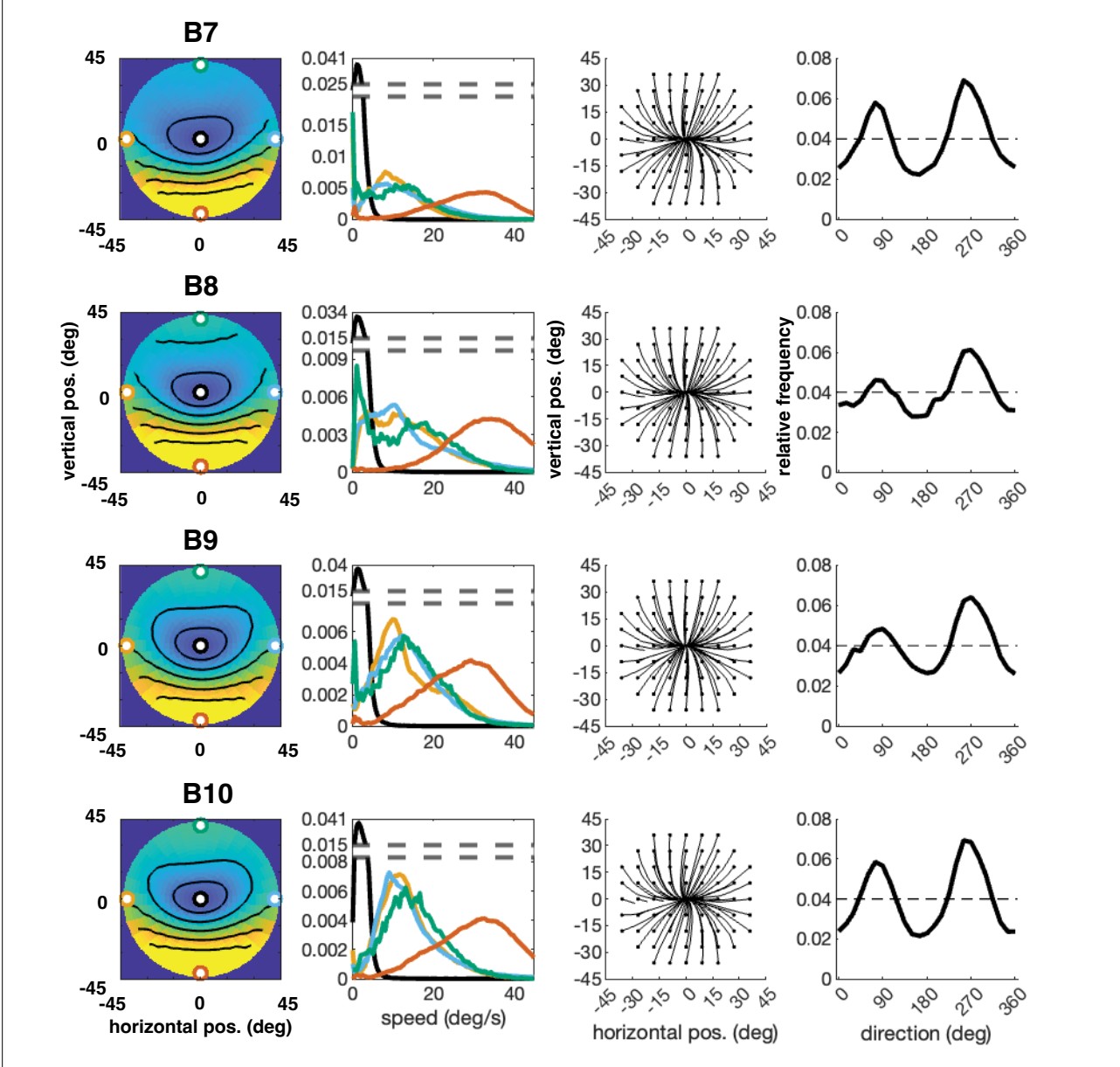

**Figure 16.** Individual participant data – compare to *Figure 4* (five participants: B6–B10). Column 1: average speed of retinal motion signals as a function of retinal position. Speed is color mapped (blue = slow, red = fast). The average is computed across all subjects and terrain types. Speed is computed in degrees of visual angle per second. Column 2: speed distributions at five points in the visual field (the fovea and four cardinal locations). Column 3: average retinal flow pattern. Column 4 Histogram of the average retinal motion directions (in Column 3) as a function of polar angle. Note: B6's data is absent because the angle of head camera during data collection precluded proper estimates of retinal flow across the visual field and resulted in poor terrain reconstruction.

in *Figure 9*. Photogrammetry can be sensitive to lighting, reflections, and other optical artifacts, but most of our recordings were done during brighter times of the day. Reconstruction in the vicinity of where subjects were walking was almost always good and intact. For one subject, certain portions of the recording had to be excluded due to poor reconstruction of the terrain resulting from bad lighting.

## Retinal motion

For retinal motion, first a set of 62,500 (produced with a method that uses a 250 × 250) vectors in eye relative coordinates are computed. These vectors tile visual space going from eccentricity 0–45 deg and cover 360 deg of polar angle. These vectors encode the position in visual space of a pixel

at a particular position in the original 250 × 250 grid. The virtual camera described in 'Optic flow estimation' is modeled as a perspective camera with 250 × 250 pixels, with a field of view of 45 deg. Therefore, the depth values at each pixel correspond to depth at the encoded location in visual space according to the 62,500 vectors. These depth values are used to calculate where a given location in 3D space (one for each pixel) moves in the current eye relative coordinate system from the current frame of the recording to the next. This results in two 3D vectors in eye relative coordinates. This allows computation of a speed (angle between these two vectors), as well as a direction (which is computed from the vector between the two vectors in the 250 × 250 grid space). Thus, each of the 250 × 250 positions in visual space has an associated speed and direction of visual motion for each frame. So for each subject, for each frame of data, we have 62,500 different speeds and directions of movement (each at a different location in visual space). When building the histograms, we put each of these 62,500 values per frame into a histogram bin according to the bin edges described above. Once that is done, the counts are normalized such that they sum to 1.

## Combining data across subjects

Once the gaze points were identified in the 3D reconstruction, gaze angle distributions as a function of terrain were calculated for a given location and then pooled across fixations and trials for a given subject. Individual subjects plots are shown in Figure 14. Similarly, once the gaze points were identified, retinal motion vectors were calculated for a given location and then pooled across fixations and trials. This gave density distributions for speed and direction across the visual field like those shown in *Figure 4* for individual subjects. Individual subject data are shown in Figures 15 and 16. Individual subject data was averaged to create *Figure 4*. Data in *Figures 5–7* were pooled across subjects, rather than averaged, as the data were binned over large angles and the point of the figures was to reveal the way the plots change with gaze angle and terrain. *Figures 10 and 11* are also pooled over subjects (*Figures 12 and 13*).

## Retinal slip during fixations

To measure the extent of the retinal slip during a fixation, we took the gaze location in the camera image at the first fixation frame and then used optic flow vectors computed by Deepflow (*Weinzaepfel et al., 2013*) to track this initial fixation location for the duration of the fixation. This is done by indexing the optic flow vector field at the initial fixation location, measuring its displacement across the first frame pair, computing its new location in the next frame, and then measuring the flow vector at this new location. This is repeated for each frame in the fixation. The resulting trajectory is that of the initial fixation location in camera image space over the duration of the fixation. For each frame of the fixation, the actual gaze location is then compared to the current location of the initial fixation location (with the first frame excluded since this is how the initial fixation is defined). The locations are converted into 3D vectors as described in 'Optic flow estimation,' and the median angular distance between gaze location and initial fixation location is computed for each fixation.

Data from this analysis can be seen in *Figure 10*, which plots the normalized relative frequency of the retinal slip during fixations. The mode of the distribution is 0.26 deg and median was 0.83 deg. This is quite good, especially given that the width of a foothold a few steps ahead will subtend about 2 deg and it is unlikely that foot placement requires more precise specification. It is likely that the long tail of the distribution results from errors in specifying the fixations rather than failure of stabilization. In particular, some small saccades are most likely included in the fixations. Other sources of error come from the eye tracker itself. Another source of noise comes from the fact that the image-based slip calculations were done at 30 Hz. Treadmill studies have found long periods (about 1.5 s) of image stability (defined as less than 4 deg/s of slip) during slow walking, with this being decreased to 213 ms for faster walking (*Dietrich and Wuehr, 2019*). *Authié et al., 2015* also found excellent stabilization in subjects walking short paths in a laboratory setting.

Manual reintroduction of retinal slip (which our measurements suggest arise from a gain in the VOR of <1) simply results in added downward motion to the entire visual field, whose magnitude is equivalent to the slip. This can be seen in *Figure 11*. Taking a median value for the slip during a 250 ms fixation of approximately 0.8 deg, the added retinal velocity would be 3.2 deg/s at the fovea.

We computed how retinal slip of this magnitude would affect speed distributions across the retina. The flow fields in the two cases (perfect stabilization and 0.8 deg of slip) are shown on the left of the

figure, and the speed distributions are shown on the right. The bottom plot shows a heat map of the difference. The added slip also has the effect of slightly shifting structure in the motion pattern upward, by however far from the fovea the eccentric location with the same speed as the slip is. When considering the average signal, this shifts the zero point upward to 4 deg of eccentricity for 4 deg/s of retinal slip. The change in speed is quite small, and the other structural features of the signal are conserved (including the radially asymmetric eccentricity versus speed gradient and the variation with gaze angle).

## Motion generated by saccades

We have thus far considered only the motion patterns generated during the periods of stable gaze since this is the period when useful visual information is extracted from the image. For completeness, we also examine the retinal motion patterns generated by saccades since this motion is incident on the retina, and it is not entirely clear how these signals are dealt with in the cortical hierarchy.

First, we show the distribution of saccade velocities and directions in *Figure 12*. Saccade velocities are generally less than 150 deg/s as might be expected from the high frequency of small movements to upcoming foothold locations. The higher velocities are likely generated as the walkers saccade from near to far, and vice versa. The direction distribution shows the over-representation of upward and downward saccades. The upward saccades most likely reflect gaze changes toward new foothold further along the path. Some of the downward saccades are from nearby locations to closer ones when more information is needed for stepping.

The effect of saccades on retinal image velocities is shown in *Figure 13*. The speed of the saccade adds to an instantaneous motion input and shifts motion directions in the opposite direction of the saccade. The figure shows retinal speed distributions at the five locations across the visual field shown in the previous figures. The top panel shows the distributions previously described for the periods of stabilization. The middle panel shows retinal speed distributions for the combined data. This shows that the speeds added by the saccades add a high-speed lobe to the data without affecting the pattern during periods of stabilization to any great extent. The lower peak resulting from the saccades results from the fact that saccades are mostly of short duration and so account for a lesser fraction of the total data. As expected, the saccades contribute high speeds to the retinal image motion that are largely outside the domain of the retinal speeds resulting from self-motion in the presence of stabilization.

## Acknowledgements

This work was supported by NIH grants R01 EY05729, T32 LM012414, and U01 NS116377.

## Additional information

### Funding

| Funder | Grant reference number | Author |
| --- | --- | --- |
| National Eye Institute | R01 EY05729 | Mary Hayhoe |
| National Institutes of Health | U01 NS116377 | Alex C Huk |
| National Institutes of Health | T32 LM012414 | Karl S Muller |
| Simons Foundation | Society of Fellows Junior Fellow | Kathryn Bonnen |
| ARVO / VSS | Fellowship | Kathryn Bonnen |

The funders had no role in study design, data collection and interpretation, or the decision to submit the work for publication.

## Author contributions

Karl S Muller, Conceptualization, Data curation, Software, Formal analysis, Supervision, Validation, Investigation, Visualization, Methodology, Writing - original draft; Jonathan Matthis, Conceptualization, Data curation, Software, Supervision, Visualization, Methodology, Writing – review and editing; Kathryn Bonnen, Conceptualization, Data curation, Methodology, Software, Visualization, Writing – review and editing; Lawrence K Cormack, Conceptualization, Formal analysis, Writing – review and editing; Alex C Huk, Conceptualization, Resources, Formal analysis, Supervision, Funding acquisition, Writing – review and editing; Mary Hayhoe, Conceptualization, Resources, Data curation, Software, Formal analysis, Supervision, Funding acquisition, Validation, Investigation, Visualization, Methodology, Writing - original draft, Writing – review and editing

## Author ORCIDs

Karl S Muller (ID) http://orcid.org/0000-0003-1319-9293
Jonathan Matthis (ID) http://orcid.org/0000-0003-3683-646X
Kathryn Bonnen (ID) http://orcid.org/0000-0002-9210-8275
Lawrence K Cormack (ID) http://orcid.org/0000-0002-3958-781X
Alex C Huk (ID) http://orcid.org/0000-0003-1430-6935
Mary Hayhoe (ID) http://orcid.org/0000-0002-6671-5207

## Ethics

Human subjects: Informed consent and consent to publish was obtained and protocols were approved by the institutional IRBs at University of Texas Austin approval number 2006-06-0085 and UC Berkeley approval number 2011-07-3429.

## Decision letter and Author response

Decision letter https://doi.org/10.7554/eLife.82410.sa1
Author response https://doi.org/10.7554/eLife.82410.sa2

---

# Additional files

## Supplementary files

• MDAR checklist

## Data availability

Processed data used to generate figures is shared via Dryad. Code to generate figures is shared via Zenodo. Raw data as well as analysis and visualization is available via Dryad.

The following datasets were generated:

| Author(s) | Year | Dataset title | Dataset URL | Database and Identifier |
|---|---|---|---|---|
| Hayhoe M | 2023 | Behavior shapes retinal motion statistics during natural locomotion | https://doi.org/10.5061/dryad.zcrjdfngp | Dryad Digital Repository, 10.5061/dryad.zcrjdfngp |
| Hayhoe M | 2023 | Behavior shapes retinal motion statistics during natural locomotion | https://doi.org/10.5281/zenodo.7055548 | Zenodo, 10.5281/zenodo.7055548 |

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
