## [Editor Report]

This important study provides new information about the statistics of "retinal" motion patterns generated by human participants physically walking a straight path in real terrains that differ in ruggedness. State-of-the-art eye, head and body tracking allowed simultaneous assessment of eye movements, head movements and gait. Compelling evidence was provided for an asymmetrical gradient of flow speeds during the gait cycle of walking, tied predominantly to vertical gaze angle, together with a radial motion direction distribution tied mostly to horizontal gaze angle. This work, by describing fundamental properties of human visual motion statistics during natural behavior, should be of great interest to scientists who seek to understand the neural computations performed by walking humans, given certain behavioral goals.

---

## [Decision Letter]

**Decision letter after peer review:**

Thank you for submitting your article "Behavior shapes retinal motion statistics during natural locomotion" for consideration by *eLife*. Your article has been reviewed by 3 peer reviewers, one of whom is a member of our Board of Reviewing Editors, and the evaluation has been overseen by Tirin Moore as the Senior Editor. The following individual involved in the review of your submission has agreed to reveal their identity: Otto Lappi (Reviewer #3).

Essential revisions:

Included here is a brief evaluation summary and list of revisions the reviewers and review editor deem essential for the authors to address. The public summaries and full, individual reviewers' recommendations for the authors are also appended below. The authors are advised to address the public summaries briefly, and the individual recommendations in a detailed, point-by-point manner.

As you will be able to read below, reviewers appreciated the importance of the study and its potentially broad interest. It uses state-of-the-art technology and presents strong behavioral evidence that should prove useful to those who seek to understand active vision and model it. The writing was relatively clear, the figures appropriate and importantly, the study design and analyses were deemed rigorous and generally appropriate. However, reviewers raised concerns with regard to some of the claims made, particularly as pertains to the novelty of the retinal flow patterns described, their implications for neural processing, and the need for additional methodological details. The key points that need to be addressed can be summarized as follows:

1. The concept that walkers stabilize their point of gaze (i.e., they "fixate" stable points on the ground ahead) is not entirely new (see references below). The fact that this generates low foveal, and higher peripheral flow velocities, especially in the lower retinal hemifield, follows simply from optical geometry. Please incorporate and discuss the references below, explaining in more quantitative and/or empirical terms, what specific gap the new methodology described in the paper actually fills:

Imai, T., Moore, S. T., Raphan, T. and Cohen, B. Interaction of the body, head, and eyes during walking and turning. Experimental Brain Research 136, 1-18 (2001).

Grasso, R., Glasauer, S., Takei, Y. and Berthoz, A. The predictive brain: anticipatory control of head direction for the steering of locomotion. NeuroReport 7, 1170-1174 (1996).

Grasso, R., Prévost, P., Ivanenko, Y. P. and Berthoz, A. Eye-head coordination for the steering of locomotion in humans: an anticipatory synergy. Neuroscience Letters 253, 115-118 (1998).

Authié, C. N., Hilt, P. M., Nguyen, S., Berthoz, A. and Bennequin, D. Differences in gaze anticipation for locomotion with and without vision. Frontiers in Human Neuroscience 9, https://doi.org/10.3389/fnhum.2015.00312 (2015).

2. Please add a discussion of the neural processing implications of your findings, preferably in the Discussion (rather than the Introduction). This could include speculations on direction tuning in cortical representations for foveal versus peripheral, upper versus the lower visual field, processing in VOR versus other gaze-stabilization circuits, etc… Feel free to invoke efficient coding or other neuro-mechanistic hypotheses as relevant.

3. All 3 reviewers commented on the need for additional methodological details – please address this fully and carefully (see details in individual reviews below).

*Reviewer #1 (Recommendations for the authors):*

While the global claims of this paper are potentially very interesting and useful, they would be strengthened by additional methodological details and consideration:

1. In some cases, it is unclear if the data shown come from a single observer (e.g., Figure 2 and Figure 3) or multiple observers? If from multiple observers, was the data based on all 11 observers or just a subset – and how was it combined across observers?

2. About half the participants were female. Were there any systematic differences in gait and related retinal flow distributions between male and female participants in the different terrains shown? I believe this study has an excellent opportunity to examine sex as a biological variable in this phenomenon – they should take advantage of it.

Additionally, the paper would be enhanced by a discussion of the following points:

1. Many figures in this paper include representations of average speed and average direction on "the retina". In all cases, a single representation is shown, as though for a cyclopean eye. However, it is unclear how this information is actually represented on the two retinas, and how information from the two eyes is then combined to a different effect (i.e. behavior) by different portions of the visual circuitry. Can the authors please discuss?

*Reviewer #2 (Recommendations for the authors):*

The overall setup is impressive, but I am curious about the computer vision component. As I understand, the optic flow statistics were calculated on a reprojected virtual scene. While the authors report an impressively small error between terrain reconstructions, I would have appreciated a Methods figure that shows example reconstructions. I am especially curious about some of the more rocky terrains, which I expect to be challenging to reconstruct with all its intricate 3D information (e.g., all the twigs and rocks sticking out of the background). Would mispredictions of the depth of these objects affect the optic flow estimations much? Also, how did lighting affect the reconstructions?

I think the implications of the findings for the neural code of vision could be presented/discussed more rigorously. For instance, the authors report several statistical peculiarities (e.g., the asymmetry between upper and lower visual fields, compression of directions, etc.). Are any of these consistent with the primate neuroscience literature? The few examples mentioned seem to (at least somewhat) contradict existing findings, except for the motion prior centered on 0. For instance, Guo et al. would suggest an overrepresentation of lateral headings, which I believe is counter to Figure 6.

Related, the comparison of the current findings to the Gibsonian view should be expanded upon. I understand that the underlying reason is different (stabilization vs. coincidence of gaze with the direction of travel), but the end result seems to be the same – that on average you get an expanding flow field. How does this finding relate to earlier work [13], which argued that the Gibsonian view rarely appears in the wild (unless you are a self-driving car)?

*Reviewer #3 (Recommendations for the authors):*

The title of the manuscript is "Behavior shapes retinal motion statistics during natural locomotion". The statement that behavior shapes retinal motion is quite obvious and known. From geometric optics and the fact that the eyes move, this statement follows. The abstract promises "the properties of the resulting retinal motion patterns".

So, any originality hinges on the word "statistics". Meaning that you expect the Results to emphasize statistics. Descriptive statistics (parameters and their distributions) you present, although some of it is perhaps more visualization than statistics, but that's okay.

Technical comments:

4.2. p3 2nd para from bottom: "Stabilization is most likely accomplished by the vestibular ocular reflex, although other eye movement systems might also be involved"; I can see no compelling reason why the compensatory eye movements would be – physiologically speaking – predominantly VOR based, as opposed to optokinetic response or smooth pursuit. I.e. why you would consider VOR "most likely". The reference you give, as far as I can tell, gives no data that directly speaks on this. You may have some argument in mind but it is not 100% clear to me what that is.

---

## [Author Response]

Essential revisions:Included here is a brief evaluation summary and list of revisions the reviewers and review editor deem essential for the authors to address. The public summaries and full, individual reviewers' recommendations for the authors are also appended below. The authors are advised to address the public summaries briefly, and the individual recommendations in a detailed, point-by-point manner.As you will be able to read below, reviewers appreciated the importance of the study and its potentially broad interest. It uses state-of-the-art technology and presents strong behavioral evidence that should prove useful to those who seek to understand active vision and model it. The writing was relatively clear, the figures appropriate and importantly, the study design and analyses were deemed rigorous and generally appropriate. However, reviewers raised concerns with regard to some of the claims made, particularly as pertains to the novelty of the retinal flow patterns described, their implications for neural processing, and the need for additional methodological details. The key points that need to be addressed can be summarized as follows:1. The concept that walkers stabilize their point of gaze (i.e., they "fixate" stable points on the ground ahead) is not entirely new (see references below). The fact that this generates low foveal, and higher peripheral flow velocities, especially in the lower retinal hemifield, follows simply from optical geometry. Please incorporate and discuss the references below, explaining in more quantitative and/or empirical terms, what specific gap the new methodology described in the paper actually fills:Imai, T., Moore, S. T., Raphan, T. and Cohen, B. Interaction of the body, head, and eyes during walking and turning. Experimental Brain Research 136, 1-18 (2001).Grasso, R., Glasauer, S., Takei, Y. and Berthoz, A. The predictive brain: anticipatory control of head direction for the steering of locomotion. NeuroReport 7, 1170-1174 (1996).Grasso, R., Prévost, P., Ivanenko, Y. P. and Berthoz, A. Eye-head coordination for the steering of locomotion in humans: an anticipatory synergy. Neuroscience Letters 253, 115-118 (1998).Authié, C. N., Hilt, P. M., Nguyen, S., Berthoz, A. and Bennequin, D. Differences in gaze anticipation for locomotion with and without vision. Frontiers in Human Neuroscience 9, https://doi.org/10.3389/fnhum.2015.00312 (2015).

We certainly do not mean to claim that stabilizing gaze is novel, and agree that the general patterns follow directly from the geometry as worked out very elegantly by Koenderink and others. We spend time describing the gaze behavior because it is essential for understanding the paper. We do not claim that the basic saccade/stabilize/saccade behavior is novel and now make this clearer. What is novel here is that we calculated the time-varying retinal motion patterns generated during the gait cycle using a 3D reconstruction of the terrain. Reference to the work above has been added. We have clarified what is novel in the text and changed the previous video (Video 2) to make the retinal motion patterns clearer. We have also added a video (Video 1) the shows how it relates to the gaze pattern.

Some Background for the reviewers below:

The novelty is in the collection of gaze statistics in natural environments, the 3D terrain reconstructions, and perhaps most importantly, the measurement of retinal motion as a function of time during the gait cycle. Previous work on the geometry has not taken the gait -induced movements into account. The other novel aspect is that the motion patterns vary with gaze location which in turn varies with terrain in a way that depends on behavioral goals. So while some aspects of the general patterns are not unexpected, the quantitative values depend on the statistics of the behavior. The actual statistics require these in situ measurements, and this has not previously been done, as stated in the abstract. Another less novel aspect is the data showing that gaze stabilization in these circumstances (ie outdoor walking in rugged environments) is very good. Other investigators (including two of those listed) have looked at stabilization gain, but not outside the laboratory. All the calculations depend on this, which is why we emphasize it.

Note that while some investigators have been well aware of gait modulations, the consequences for both retinal motion and how flow is used, have not previously been measured and have been consistently underestimated. This is the primary driver of the motion statistics (lateral, vertical, and forward) a fact that has been neglected. Other details of the motion statistics such as the variation in direction densities, and the horizontal band of low velocities through the fovea have not previously been demonstrated, and simply the actual distributions of direction and speed across the retina. These details affect fundamental questions, like what an optimal stimulus for an MST receptive field might be or how self-motion and world motion are parsed by the visual system.

One of the difficulties in describing our work is that there are different groups of researchers who are sensitive to different aspects of the generation and properties of retinal motion, and how it might be used. For example, the neurophysiology has of necessity all been done with a stable head. This means that there is no gait cycle and no stabilizing eye movements. Investigators use pursuit eye movements typically against a background whose motion is not time-varying in a way linked to gait, and therefore has a different geometry. In addition, many of the motion stimuli chosen in experiments are not clearly related to patterns that might be generated during natural body movements (for example, non-zero motion at the fovea). This has also made it hard to make tight links between the measured statistics and properties of cells in the visual pathways, and we cited those we felt most directly comparable. The psychophysics community also almost universally uses seated observers and pursuit eye movements, simulating a constant body heading direction. Because the retinal motion is directly dependent on behavior and gaze stabilization during the gait cycle, we felt it necessary to make this clear, even though some might find it unremarkable.

Other groups of researchers (like those cited) especially those who look at natural behavior are sensitive to eye and body movements and the need for image stabilization, and the papers listed address important and interesting questions about coordinated whole body movements. We have cited those references to acknowledge this work on basic gaze behavior and the high gain of the stabilizing mechanisms. However, the goal of the current paper was to specify how this affects retinal motion, and we are not aware of previous work that calculates the retinal motion statistics in the natural world generated by locomotion.

The manuscript has been edited to make the novelty clearer.

2. Please add a discussion of the neural processing implications of your findings, preferably in the Discussion (rather than the Introduction). This could include speculations on direction tuning in cortical representations for foveal versus peripheral, upper versus the lower visual field, processing in VOR versus other gaze-stabilization circuits, etc… Feel free to invoke efficient coding or other neuro-mechanistic hypotheses as relevant.

The measured statistics provide a well-defined set of hypotheses about the pattern of direction and speed tuning across the visual field. We have added text (see p 14, 15) to be more explicit about what we would expect to find in MT and MSTd if those cell properties are indeed shaped by these statistics. The points of comparison in the existing literature are hard to find because the stimuli have not been closely matched to actual retinal flow patterns. This is discussed on pp 14-16. We have done what we could, but this might fall short of expectations because of difficulty finding comparable stimulus conditions.

3. All 3 reviewers commented on the need for additional methodological details – please address this fully and carefully (see details in individual reviews below).

A more detailed description of the methods including the photogrammetry and the reference frames for the measurements has been added primarily to the Methods section.

Figure 1 is an example segment of a record for a single subject. All other Figures are data averaged across all subjects as described in the Methods.

Standard errors between Subjects have now been added to Figure 2, 3 and 4 b and d and Figure 12. The effects of horizontal and vertical gaze angle and terrain are for demonstration so we have not calculated the standard errors for those Figures (see above).

Reviewer #1 (Recommendations for the authors):While the global claims of this paper are potentially very interesting and useful, they would be strengthened by additional methodological details and consideration:1. In some cases, it is unclear if the data shown come from a single observer (e.g., Figure 2 and Figure 3) or multiple observers? If from multiple observers, was the data based on all 11 observers or just a subset – and how was it combined across observers?

This has been clarified in Figure legends and Methods. Figure 1 is an example segment of a record for a single subject. All other Figures are data averaged across all subjects as described in the Methods. Standard errors between Subjects have now been added to Figure 2, 3 and 4 b and d and Figure 12. The effects of horizontal and vertical gaze angle and terrain are for demonstration so we have not calculated the standard errors for those Figures (see above).

Were there any systematic differences in gait and related retinal flow distributions between male and female participants in the different terrains shown? Likewise, age and optical correction may have an impact on gait, head, and eye movements. Neither is indicated in the paper. This should be remedied and the authors should include a discussion of their likely impact on the behaviors examined.

As described above and in the text, many factors influence step length, step speed, and linked to this, gaze location. Individual subjects also differ in tradeoffs between energetic costs and stability. Given this, the most likely effect of sex is leg length. Since we were most concerned to describe properties that are generally true over a range of natural circumstances we felt that specific investigation of sex was outside the scope, and was only one of many factors that will modulate retinal motion. We all spend a lot of our time walking over ground planes and our goal was to capture a valid basis for how that might relate to the structure of motion processing in the brain. In other work we are examining individual sensitivity to their own motion patterns.We have now indicated the nature of the variability between subjects and show individual subject data in the Supplementary Materials. We agree these are important questions and worth investigating. However, the goal of the present study was to specify the common features for humans walking over ground planes, so we chose not to focus on the individual variability but rather on the commonalities. The properties of the statistics that we report are true for all subjects and should allow one to evaluate the kinds of changes occasioned by looking closer to the body, for example. By showing how the retinal motion derives from gait modulations, gaze angle, and terrain and how these factors influence the statistics we hope that in principle it should be possible to estimate the consequences of increased blur for example, which would lead to fixations closer to the body.

Additionally, the paper would be enhanced by a discussion of the following points:1. Many figures in this paper include representations of average speed and average direction on "the retina". In all cases, a single representation is shown, as though for a cyclopean eye. However, it is unclear how this information is actually represented on the two retinas, and how information from the two eyes is then combined to a different effect (i.e. behavior) by different portions of the visual circuitry. Can the authors please discuss?

The eye tracker combines the estimates of gaze point from the two eyes (mean location) and that is what we use as the point of fixation. We estimate retinal motion on a single cyclopean retina fixated on that point. It would be nice to have eye specific calculations but the intrinsic error in the measurements make this of limited value given the current accuracy of the trackers. This is now specified in the Methods Section.

Reviewer #2 (Recommendations for the authors):The overall setup is impressive, but I am curious about the computer vision component. As I understand, the optic flow statistics were calculated on a reprojected virtual scene. While the authors report an impressively small error between terrain reconstructions, I would have appreciated a Methods figure that shows example reconstructions. I am especially curious about some of the more rocky terrains, which I expect to be challenging to reconstruct with all its intricate 3D information (e.g., all the twigs and rocks sticking out of the background). Would mispredictions of the depth of these objects affect the optic flow estimations much? Also, how did lighting affect the reconstructions?

An example is now shown in Figure 9. For one subject, certain portions of the recording had to be excluded due to poor reconstruction of the terrain resulting from bad lighting. The reconstructions can be sensitive to lighting but most of our recordings were done during brighter times of the day. Reconstruction in the vicinity of where subjects were walking was almost always good and intact.

I think the implications of the findings for the neural code of vision could be presented/discussed more rigorously.

We have pointed out a number of contradictions in the Discussion. This is one of the reasons we feel that we are limited in what we can say about the neurophysiology.

Related, the comparison of the current findings to the Gibsonian view should be expanded upon. I understand that the underlying reason is different (stabilization vs. coincidence of gaze with the direction of travel), but the end result seems to be the same – that on average you get an expanding flow field. How does this finding relate to earlier work [13], which argued that the Gibsonian view rarely appears in the wild (unless you are a self-driving car)?

Our previous paper (Matthis et al. 2022) goes into this issue in depth, and we have added discussion of this in the present paper to make it clear that retinal motion is generated not directly by body motion, but indirectly by stabilizing rotations and translations of the eye to counteract body motion during a step. See page 12 and the videos. The implications are for the way the motion is used, and this is related but a bit orthogonal to the current ms. The Matthis et al. paper measures the time-varying nature of the retinal flow field through the gait cycle. What this means is that the direction of the head in space varies wildly during a step, so unless there is a mechanism to integrate heading direction over two steps there is no way that heading can be used to steer towards a goal as is commonly thought. This theme has been a common one in both psychophysics and neurophysiology in the past. Since the implication is that we have as a field been posing the wrong question, as you can imagine it has met with some resistance so we did not seek to highlight it here, since in the current ms, statistics are integrated over time, terrains, subjects etc and the time-varying nature of the flow was not the focus.

Reviewer #3 (Recommendations for the authors):

Technical comments:4.2. p3 2nd para from bottom: "Stabilization is most likely accomplished by the vestibular ocular reflex, although other eye movement systems might also be involved"; I can see no compelling reason why the compensatory eye movements would be – physiologically speaking – predominantly VOR based, as opposed to optokinetic response or smooth pursuit. I.e. why you would consider VOR "most likely". The reference you give, as far as I can tell, gives no data that directly speaks on this. You may have some argument in mind but it is not 100% clear to me what that is.

In the literature on visual responses to optic flow, nearly all the work is with a seated observer, so there is no vestibular signal, and much of the focus has been on the role of optic flow in heading towards a goal. In this literature investigators typically use a pursuit eye movement target. We use this language because in natural behavior the head is always moving and the role of the VOR is to stabilize the image on the retina. The head oscillations during locomotion generate the acceleration signals that drive the VOR. The only downside to the VOR here is the low temporal frequency loss of sensitivity which might need to be compensated for perhaps by some predictive mechanism when moving at a constant velociity (because OKN and pursuit are longer latency than the VOR – approximately 100 msec versus 10 msec). Since how it is done is tangential to the goals of the paper and we can’t resolve it, we used what we hoped was diplomatic language. See Land (2019) for a discussion of the ubiquity of the VOR in the animal kingdom. Note that damage to the vestibular system is highly debilitating. We have added to the discussion of stabilization mechanisms on p13 and 18.